# Autoinhibition and activation of myosin VI revealed by its cryo-EM structure

Fengfeng Niu [1,2,6], Lingxuan Li [1,6], Lei Wang[1], Jinman Xiao[3,4], Shun Xu[1], Yong Liu [1], Leishu Lin [1], Cong Yu [3,4,5] ✉ & Zhiyi Wei [1,2,5] ✉

Myosin VI is the only molecular motor that moves towards the minus end along actin filaments. Numerous cellular processes require myosin VI and tight regulations of the motor's activity. Defects in myosin VI activity are known to cause genetic diseases such as deafness and cardiomyopathy. However, the molecular mechanisms underlying the activity regulation of myosin VI remain elusive. Here, we determined the high-resolution cryo-electron microscopic structure of myosin VI in its autoinhibited state. Our structure reveals that autoinhibited myosin VI adopts a compact, monomeric conformation via extensive interactions between the head and tail domains, orchestrated by an elongated single-α-helix region resembling a "spine". This autoinhibited structure effectively blocks cargo binding sites and represses the motor's ATPase activity. Certain cargo adaptors such as GIPC can release multiple inhibitory interactions and promote motor activity, pointing to a cargo-mediated activation of the processive motor. Moreover, our structural findings allow rationalization of disease-associated mutations in myosin VI. Beyond the activity regulation mechanisms of myosin VI, our study also sheds lights on how activities of other myosin motors such as myosin VII and X might be regulated.

The myosin superfamily encompasses a vast array of molecular motors that move along actin filaments and serve diversely fundamental functions, spanning from muscle contraction to intracellular transport[1–3]. These motors consist of three essential regions: the N-terminal motor domain (MD) is responsible for ATP hydrolysis and actin attachment, allowing the myosin motor to generate movement along actin filaments; the varying numbers of IQ motifs interact with light chain proteins, such as calmodulin (CaM), forming a rigid lever arm (LA) structure that supports movement size; and the C-terminal tail is highly diverse among different myosins and plays roles in cargo recognition and the regulation of motor activity[4–6]. However, the interplay among these three regions in myosins that governs motor behavior is still largely unknown.

Myosin VI was characterized as the only myosin walking towards the minus ends of actin filaments, due to the unique sequence of Insert2 (In2) connecting the MD and LA in myosin VI that drives the LA movement towards the reverse direction in contract to other myosins[7,8]. This exceptional directionality endows myosin VI a pivotal role in a wide range of cellular activities, including vesicle transport, endocytosis, synaptic plasticity, autophagy, and cell motility[9–11]. Dysfunctions in myosin VI have been associated with various human diseases, such as auditory and vestibular defects, familial cardiomyopathy, and certain cancers[12,13]. The multifunctionality of

[1]Department of Neuroscience and Brain Research Center, School of Life Sciences, Southern University of Science and Technology, Shenzhen, Guangdong, China. [2]Shenzhen Key Laboratory of Biomolecular Assembling and Regulation, Shenzhen, Guangdong, China. [3]Department of Chemical Biology, School of Life Sciences, Southern University of Science and Technology, Shenzhen, Guangdong, China. [4]Guangdong Provincial Key Laboratory of Cell Microenvironment and Disease Research, and Shenzhen Key Laboratory of Cell Microenvironment, Shenzhen, Guangdong, China. [5]Institute for Biological Electron Microscopy, Southern University of Science and Technology, Shenzhen, Guangdong, China. [6]These authors contributed equally: Fengfeng Niu, Lingxuan Li. ✉e-mail: yuc@sustech.edu.cn; weizy@sustech.edu.cn

myosin VI requires its remarkable capacity to interact with a diverse array of cargo adapter proteins, such as endocytic adapters GIPC, clathrin light chain a (CLCa), Dab2, and TOM1, autophagy receptors NDP52, TAX1BP1, and Optineurin, as well as ubiquitin proteins, through the C-terminal cargo-binding domains (CBD1 and CBD2) of myosin VI (Fig. 1a)[14].

In the absence of cargos, many myosins (e.g., myosin II, V, VI, VII, and X) employ autoinhibition as a regulatory mechanism[15-25]. Autoinhibition keeps the motor activity suppressed until specific signals or stimuli are present to trigger activation. Recent cryogenic electron microscopy (cryo-EM) studies have provided structural insights into the autoinhibited architectures of myosin II and V, elucidating how

their tail regions interact with the MDs to form a compact, closed conformation[26-30]. Previous biochemical investigations on myosin VI have also suggested that its tail region folds back onto the MD, obstructing actin-binding and ATPase activity[21,22]. However, unlike myosin II and V, which contain long coiled coils in the tail regions for dimer formation, myosin VI possesses a single-α-helix (SAH) and exists as a monomer in the autoinhibited state[21,22]. Due to lack of comprehensive structural characterization, the molecular basis underlying the transition of myosin VI between its autoinhibited and active forms remains unclear.

In this study, we solved the cryo-EM structure of myosin VI in its autoinhibited state. The structural analysis uncovered the unexpected

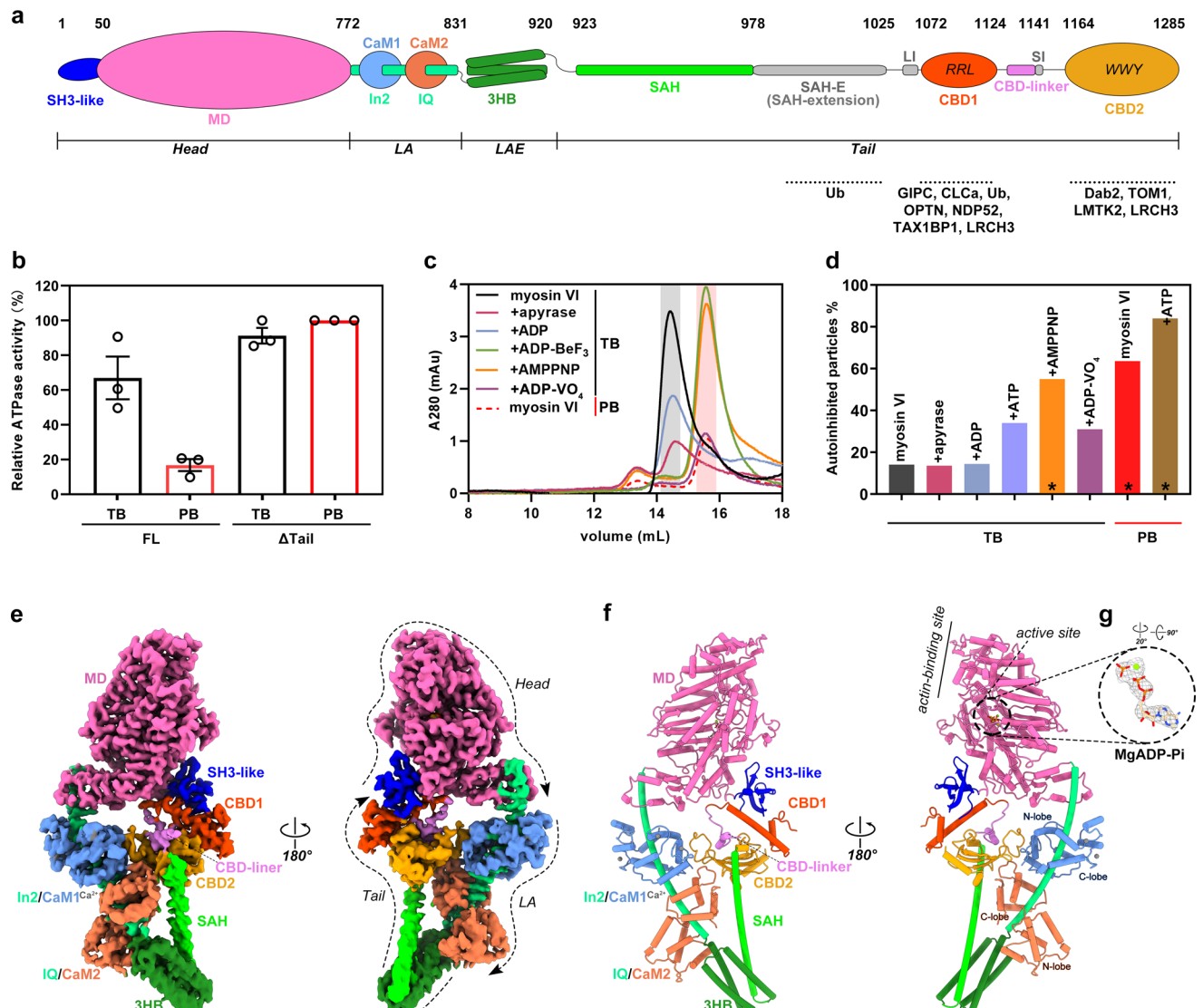

**Fig. 1 | Overall structure of myosin VI in the autoinhibited state. a** Schematic domain organization of myosin VI. Two CaM molecules are depicted binding to specific regions in myosin VI. Two essential cargo-binding motifs of RRL and WWY in CBD1 and CBD2, respectively, are highlighted with their corresponding cargos/cargo adapters indicated. The color code used here is applied throughout the manuscript. **b** ATPase activities of myosin VI and its tail-deleted mutant measured in TB and PB. The measured activity of the ΔTail mutant in PB is used to indicate 100 % activation of myosin VI and other samples are normalized to it for comparison. All the reactions were experimentally repeated three times and all data are presented as mean values ± SEM in this and the following measurements of ATPase activity. **c** aSEC profiles of myosin VI samples in different nucleotide and buffer conditions. The elution volume for autoinhibited and open myosin VI is indicated with red and gray color,

respectively. The samples in TB were prepared either without additive or with an additional 8 U/mL apyrase, 1 mM ADP/ATP, or 1 mM ATP analogs. The samples in PB were prepared by exchanging buffer from TB to PB. **d** Percentage of autoinhibited particles of myosin VI in indicated conditions analyzed by cryo-EM. The samples were prepared with similar treatments in panel (**c**) and three samples marked with stars were used for cryo-EM structure determination. **e** Cryo-EM map of myosin VI in the autoinhibited state, highlighting its domains and two CaM molecules. The overall extension direction of the head, LA, LAE, and tail regions is indicated. Map contour level = 0.1. **f** Atomic model of autoinhibited myosin VI. The active site and actin-binding site in the MD and the N/C-lobe in the CaM molecules are labeled. Four Ca²⁺ ions binding to CaM1 are depicted as gray spheres. **g** Density map of MgADP-Pi at the active site, with its structure overlapped. Map contour level = 0.1.

interactions among the three regions in myosin VI that maintain the inhibitory conformation. These extensive interactions not only effectively block all identified cargo-binding sites within the CBDs, but also restrict the swing of the LA, which is required for ATP turnover of the MD. Surprisingly, we found that the extension of SAH (SAH-E) in the tail region interacts with the active site of the MD, directly interfering with the enzyme activity of myosin VI. Together with the SAH, this simple helical region plays a profound role in coordinating the head, LA, LAE, and CBDs to maintain the autoinhibited state of myosin VI. Furthermore, we discovered that GIPC and CLCa can disrupt multiple interactions between CBD1 and other domains in myosin VI to release its autoinhibition, allowing myosin VI to be activated and further dimerized for cargo transport. Collectively, our research employs a combination of biochemical and structural analyses to unravel the intricacies of the regulation of myosin VI activity. Finally, by comparing the autoinhibited structure of myosin VI with those of myosin II and V, we provide insights into the broader understanding of molecular motor proteins.

## Results

### Phosphate binding promotes the autoinhibited state formation of myosin VI

Human myosin VI protein was purified in complex with its light chain CaM (Fig. 1a and Supplementary Fig. 1a). Interestingly, analytical size exclusion column (aSEC) analysis revealed distinct conformations of myosin VI in the tris buffer (TB) and phosphate buffer (PB), as indicated by their different elution volumes (Supplementary Fig. 1a). The myosin VI samples in these two buffer conditions were then analyzed by using negative-stain EM. In TB, myosin VI particles exhibited a bent structure (Supplementary Fig. 1b), while in PB, myosin VI adopted a relatively straight shape (Supplementary Fig. 1c), resembling the autoinhibited state observed previously[21,31]. These findings suggest that the PB condition promotes the formation of the autoinhibited state. This notion is supported by the measured ATPase activity of myosin VI in the two conditions. The ATPase activity was approximately 4-fold lower in PB compared to TB (Fig. 1b), comparable to the activity difference of 5–7 folds observed in other myosins between their autoinhibited and active states[19,25].

Consistent with the previous observations that ATP promotes the formation of the autoinhibited conformation[21,32], the addition of ATP analogs in TB led to an elution volume shift of myosin VI in aSEC analysis, resembling the change observed when switching the buffer condition from TB to PB (Fig. 1c). However, the addition of ADP or the enzyme apyrase, which hydrolyzes ATP and ADP, did not induce any change in elution volume (Fig. 1c). Given the similar change observed in PB containing abundant phosphates, it is likely that the binding of phosphate or the γ-phosphoryl (γ-Pi) group in ATP to myosin VI triggers the autoinhibition formation. Consistently, in our EM analysis, the percentage of autoinhibited particles in the TB condition increased ~2–3 fold with the addition of ATP or its analogs (Fig. 1d).

### Overall structure of myosin VI in the autoinhibited state

To uncover the structural basis of myosin VI autoinhibition, we selected myosin VI samples under three conditions with the high percentage of autoinhibited particles for cryo-EM single-particle analysis (Fig. 1d and Supplementary Figs. 2–4). Consistent with our biochemical findings, myosin VI in these conditions has essentially the same shape (Supplementary Fig. 5a, b). By combinatorial usage of the three datasets, we determined the cryo-EM structure of autoinhibited myosin VI at an overall resolution of 3.54 Å (Fig. 1e, Supplementary Fig. 6a–d, Supplementary Table 1 and Supplementary Movie 1). All the defined structural elements, except for a few flexible linker sequences, can be confidently assigned without ambiguity (Fig. 1f, Supplementary Figs. 7 and 8). In line with the critical role of Pi binding in the autoinhibition of myosin VI, a similar patch of density was observed at the canonical γ-Pi

binding site within the active site of myosin VI in all three conditions (Supplementary Fig. 5c). Notably, the EM density in the PB + ATP condition clearly indicates the presence of ADP·Pi at the active site (Supplementary Fig. 4d). Considering that this condition promotes the highest level of autoinhibited state formation (Fig. 1d), our final model incorporates ADP·Pi at the active site (Fig. 1g).

In the autoinhibited structure, the monomeric myosin VI in complex with two CaM molecules forms a unique "head-to-tail" architecture (Fig. 1e, f), in which myosin VI ties its "head" with the "tail" through the interaction between the N-terminal SH3-like domain and CBD1 (Fig. 1e, f). The conformation of the MD in the head region closely resembles the Pre-Power Stroke (PPS) state including the actin-binding site, the active site, and the converter (Supplementary Fig. 8a)[33,34], leading to the LA and the following LA extension (LAE) consisting of a three-helix bundle (3HB) extending to the opposite side of the head (Fig. 1f). The observed PPS state of the MD in the structure explains why myosin VI in PB mainly adopts the autoinhibited conformation even in the absence of ATP or its analogs. The binding of Pi at the canonical γ-Pi binding site likely triggers a conformational change in the active site, leading to the transition of myosin VI towards the PPS state. This notion is supported by the crystal structure of the MD bound with soaked Pi, which also adopts the PPS state[33]. Meanwhile, considering the PPS state has been demonstrated in myosin field to have a weak binding ability to actin filament[35], the PPS state of the MD observed in our structure suggests that myosin VI in its autoinhibited conformation may weakly associate with actin. A sharp U-turn between the LAE and SAH in the tail makes the head/tail interaction possible by bringing the CBD region near the head (Fig. 1e, f). In addition to the head/tail interaction, extensive intermolecular/interdomain interactions are observed in the autoinhibited structure of myosin VI. Notably, these interactions involve not only all the well-characterized domain but also a previously undefined sequence between CBD1 and CBD2, named CBD-linker hereafter, which stitches the two CBDs together (Fig. 1e, f).

### Extensive intermolecular/interdomain interactions maintain the autoinhibited state of myosin VI

In the autoinhibited state of myosin VI, an intricate network of interactions is formed among different domains of myosin VI and between the tail and two CaM molecules in the LA, constraining the conformations of both the head and tail (Fig. 2), which are described in detail below.

The N-terminal SH3-like domain mediates the head/tail interaction (interface-1). The head/tail interaction is a commonly observed regulatory mechanism in many motor proteins' autoinhibited states[26,36]. Myosin VI mainly utilizes its N-terminal SH3-like domain to bind to the CBD1 (Fig. 2a), which is formed by two consecutive helices in the tail (Supplementary Fig. 8e)[37–39]. Interestingly, despite having a SH3 fold, the SH3-like domain of myosin VI does not employ the typical target-binding groove found in conventional SH3 domains to interact with the CBD1 (Supplementary Fig. 8b)[40,41]. Instead, it forms another highly hydrophobic groove between its two termini, accommodating the bulky sidechains of W1115 and Y1112 in a long helix of CBD1 (Fig. 2a). The hydrophobic interaction is further strengthened by several hydrogen bonds (Fig. 2a), ensuring a stable and specific head/tail interaction. Notably, the SH3-like domain in the conventional myosin II is important for the autoinhibition formation[27–30], while the role of the SH3-like domain in many unconventional myosins has not been elucidated. Our discovery of the SH3-like domain in the formation of myosin VI autoinhibition suggests the potential role of this domain also in regulating the activity of other unconventional myosins.

Noteworthy, the PPS state of the MD is required for the autoinhibition formation of myosin VI. By comparing the MD structures in different states, we found that the converter of MD in Rigor, Post-Rigor, and ADP states would clash with the CBD region, while the

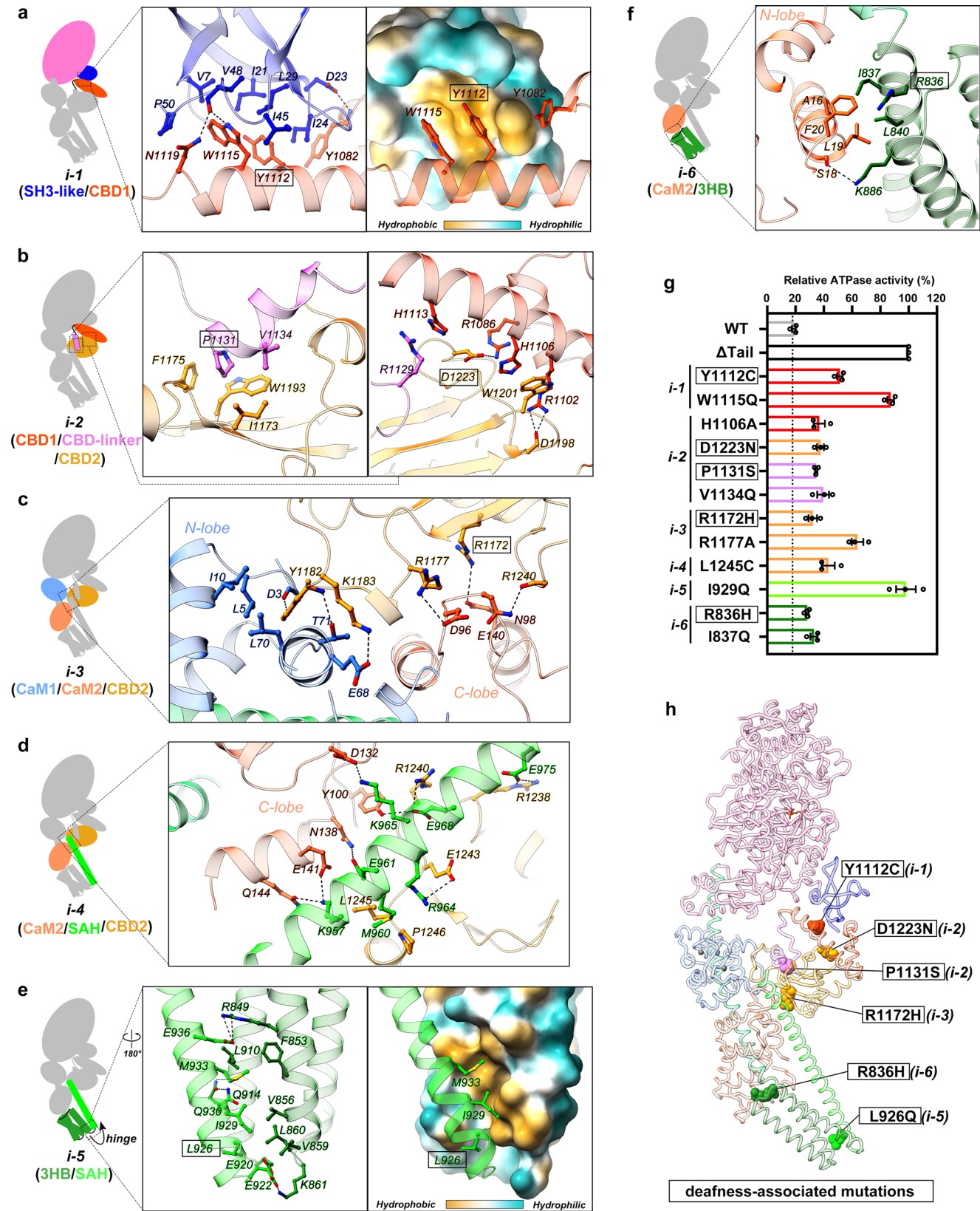

distinct orientation of the converter in the PPS state allows the close contact between the MD and CBD1 (Supplementary Fig. 9a–c). Consistent with the promotion of the PPS state of the MD in the PB condition, we observed the prominent interaction between the MD and the CBD1 in PB but not in TB using aSEC analysis (Supplementary Fig. 9d, e). Additionally, the W1115Q mutation at the head/tail interface

disrupts this binding (Supplementary Fig. 9f), confirming our structural observation.

The two CBDs are assembled as a compact structure in the autoinhibited state (interface-2). The CBD region of myosin VI plays a critical role in cargo recognition, with CBD1 and CBD2 responsible for recruiting different cargos[14]. During our model-building process, we

**Fig. 2 | Interfaces in the formation of the autoinhibited myosin VI structure.** **a–f** The six interfaces involved in the formation of autoinhibited myosin VI. Interface-1 (**a**) involves the SH3-like and CBD1 domains for the head/tail interaction; interface-2 (**b**) is formed by CBD1, the CBD-linker, and CBD2 for the assembly of the whole CBD region; interface-3 (**c**) is constituted by CaM1, CaM2 and CBD2 for the LA/CBD interaction; interface-4 (**d**) is formed by CaM2, the SAH and CBD2; interface-5 (**e**) refers to the interaction between the 3HB and SAH for the hinge formation; and interface-6 (**f**) involves CaM2 and 3HB. The hydrophobic surfaces on the SH3-like and 3HB domains in interface-1 and interface-5 are shown in panels (**a**) and (**e**), respectively. **g** ATPase activity measurements of myosin VI and its interface mutants in PB. The measured activity of the ΔTail mutant is set as 100% activation of myosin VI and other samples are normalized to it for comparison. The deafness-associated mutants are boxed. **h** Highlighted mutations associated with deafness in the interfaces of the autoinhibited myosin VI structure. These deafness-associated mutation sites are highlighted with boxed in the corresponding interfaces as shown in panels **a–c**, **e**, and **f**, respectively. The labels 'i-1' to 'i-6' in panels (**g**) and (**h**) correspond to the six essential inhibitory interfaces shown in panels (**a–f**).

found an unexpected density in the cargo-binding pocket of CBD2 after fitting all known domain structures into the EM density (Supplementary Fig. 6h). The high-quality density map allowed us to trace a peptide sequence from the C-terminus of CBD1 to the N-terminus of the small insertion (SI), an alternatively spliced region. This essential sequence, named CBD-linker, which links CBD1 and CBD2 together as a compact structure, forms a helical structure at its C-terminus to interact with the CBD2 pocket primarily through hydrophobic interactions (Fig. 2b, left panel). Meanwhile, the CBD-linker also connects to CBD1 by a cation-π interaction between $R1129^{CBD-linker}$ and $H1113^{CBD1}$ (Fig. 2b, right panel). Additionally, the two CBDs directly interact with each other via a stacking force between the aromatic sidechains of $H1106^{CBD1}$ and $W1201^{CBD2}$, along with hydrogen bonding and charge-charge interactions (Fig. 2b, right panel). These interactions collectively assemble the two CBDs and place CBD1 to the right position for its binding to the SH3-like domain in the head.

CBD2 tethers the LA and SAH to the CBD region (interface-3&4). In addition to forming a compact structure with the CBD-linker and CBD1, CBD2 is directly involved in interactions with the LA (Fig. 2c) and SAH (Fig. 2d). The LA in myosin VI is formed by two CaM molecules (CaM1 and CaM2) binding to the In2 and IQ motif, respectively (Supplementary Fig. 8c)[42]. Interestingly, the N-lobe of CaM1 and C-lobe of CaM2 form a shallow cavity to hold a long loop in the CBD2 (Fig. 2c). Remarkably, this loop, compared to its conformation in the crystal structures of CBD2[43,44], undergoes a large conformational change, leading to the reversed sidechain orientation of the residue Y1182 (Supplementary Fig. 8f). This unique orientation allows Y1182 to engage in extensive hydrophobic interactions within the cavity, which is further stabilized by a hydrogen bond formed between the hydroxyl group in Y1182 and a mainchain carbonyl oxygen in the CaM1 N-terminus (Fig. 2c). Additionally, CBD2 closely contacts with the C-lobe of CaM2, creating several highly charged surface patches to accommodate the charged residues in the C-terminal part of the SAH (Fig. 2d). This interaction ensures the proper position of the SAH extending towards the head. Hence, CBD2 serves as a critical mediator, capturing the LA and tail and stabilizing the CBD orientation for the head/tail interaction.

The LAE connects the LA and SAH for the hinge formation (interface-5&6). In our autoinhibited myosin VI, a hinge structure is found between the LAE and SAH (Fig. 2e). Specifically, the N-terminal part of SAH forms a four-helix bundle (4HB) structure with the 3HB of the LAE. Several hydrophobic residues in the SAH contribute to the hydrophobic core formation of this 4HB, stabilizing the sharp turn conformation of the hinge structure (Fig. 2e). However, the 4HB formed by the LAE and SAH is not as tightly packed as the typical 4HB structure (Supplementary Fig. 8d), likely due to a high proportion of hydrophilic residues in the SAH sequence. Meanwhile, the LAE also interacts with the LA by binding to the N-lobe of CaM2 (Fig. 2f), similar to the previous observation in the crystal structure containing the LA and LAE[42] (Supplementary Fig. 8c).

### Interface and disease mutations disrupt the autoinhibition of myosin VI

The residues forming the autoinhibitory interfaces are highly conserved across different species from fly to human (Supplementary Fig. 10), indicating that the unique "head-to-tail" architecture revealed in our autoinhibited structure is shared in class VI myosins. To further validate the important role of the interfaces in maintaining the autoinhibited state, we designed a series of disruptive interface mutations and analyzed the impacts of these mutations on ATPase activity. Consistent with our structural findings, all mutations led to increased ATPase activity compared to wild-type myosin VI (Fig. 2g and Supplementary Fig. 11). Remarkably, two specific mutations, W1115Q and I929Q, disrupting interface-1 and interface-5, respectively, boost ATPase activity to a level comparable to that of the myosin VI fragment with its tail region deleted (Fig. 2g). These findings are strongly aligned with the profound roles of the head/tail interaction and the hinge formation in maintaining the autoinhibited conformation of myosin VI.

Given the well-established linkage between mutations in myosin VI and hereditary hearing loss[12], we mapped the identified missense mutation sites onto the autoinhibited structure of myosin VI. Strikingly, our analysis revealed six deafness-associated mutations, including Y1112C[45], D1223N[46], P1131S[47], R1172H[48], R836H[49], and L926Q[50], located at the autoinhibitory interfaces (Fig. 2h). Among these mutations, D1223N, which replaces a negatively charged residue with a neutral one, presumably abolishes the charge-charge interaction with R1086 in the CBD1/CBD2 interface (Fig. 2b). The mutations, Y1112C, P1131S, and L926Q, which substitute hydrophobic residues with hydrophilic ones, may weaken the hydrophobic interactions in the different interfaces (Fig. 2a, b, e). Consistent with our structural analyses, these deafness-associated mutations led to a mild yet significant increase in the ATPase activity of myosin VI (Fig. 2g). Hence, the myosin VI structure provides a molecular basis for studying the pathological mechanisms underlying the diseases caused by myosin VI dysfunction.

### The SAH-extension interacts with the active site in the MD to promote the autoinhibitory effect

During the cryo-EM data processing of myosin VI, an intriguing subset of particles was identified, which displayed a slender protrusion, emanating from the SAH and extending towards the MD, without altering the overall conformation (Fig. 3a). Through the focused 3D classification, approximately 20,000 particles exhibiting this protrusion were isolated, enabling us to refine a density map with an overall resolution of 4.14 Å (Supplementary Fig. 6e–g and Supplementary Table 1). In this refined map, a prominent helical density emerged, with one end proximate to the C-terminus of the SAH and the other nestled within a cavity formed by the SH3-like and MD domains (Fig. 3b). We henceforth termed this helix as SAH-extension (SAH-E), which lacks a typical SAH sequence pattern rich in E, R, and K residues[51,52]. To unravel the details of the interaction between the SAH-E and MD, we employed Alphafold2-powered prediction[53–55] to guide the tracing of the SAH-E sequence, encompassing ~40 residues, into the corresponding density envelope (Fig. 3c, Supplementary Figs. 6i and 7, and Supplementary Movie 2).

The SAH-E inserts its C-terminal segment into the cavity formed by the L50 and N-term subdomains, as well as the SH3-like domain (Fig. 3d), creating an additional head/tail interaction (interface-7) for myosin VI's autoinhibition. Unlike the SH3-like/CBD2 interaction that resides distal to the active site of the MD, the SAH-E directly contacts

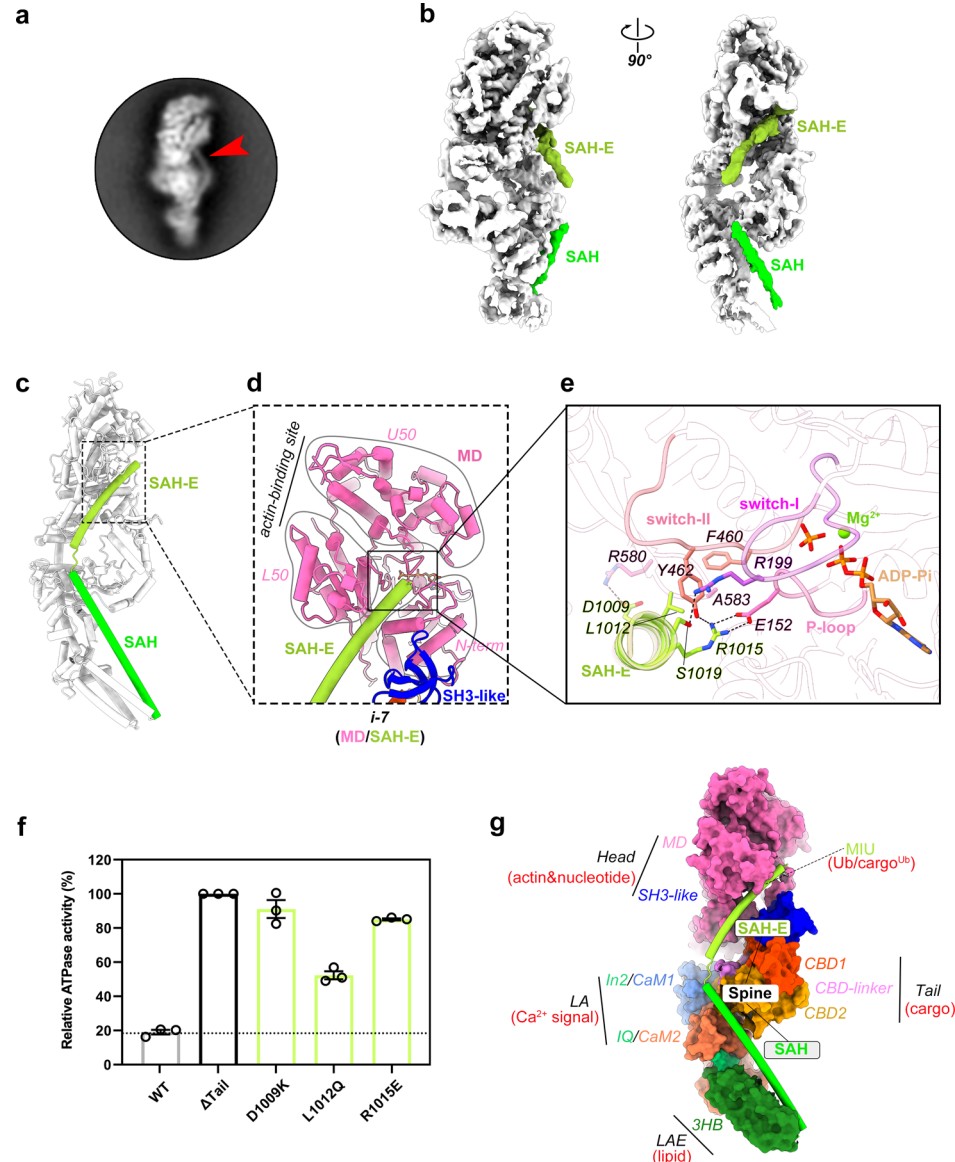

**Fig. 3 | The "spine"-like structure formed by the SAH and SAH-E in blocking the active site and integrating myosin VI domains. a** Representation of a 2D class of myosin VI particles showing a stick-like density extending from the end of SAH, indicated by a red arrowhead. **b, c** Cryo-EM map with contour level of 0.1. (**b**) and atomic model (**c**) of autoinhibited myosin VI with the SAH and SAH-E highlighted. The SAH-E helix is inserted into a cavity in the MD. **d** Interface-7 between the SAH-E and MD in autoinhibited myosin VI. The cavity is formed by the SH3-like domain and the N-term and L50 subdomains in the MD. The actin-binding site between U50 and L50 is indicated. **e** Detailed depiction of the molecular interaction at interface-7. The C-terminal of SAH-E interacts directly with essential elements (P-loop, switch-I, switch-II) of the active site in the MD. **f** In vitro ATPase activity measurements for three myosin VI mutants with the impaired SAH-E/MD interaction. The measured activity of the ΔTail mutant is set as 100 % activation of myosin VI and other samples are normalized to it for comparison. **g** Representation of the SAH and SAH-E with the surrounding regions in the autoinhibited state of myosin VI. These two consecutive single α-helices link the head, LA, LAE, and tail together. Various factors, such as ubiquitin (Ub), ubiquitinated cargo (cargo$^{Ub}$), Ca$^{2+}$, and lipids that interfere with this linkage potentially release the autoinhibition of myosin VI.

with the catalytic center of the MD (Fig. 3e), including the P-loop, switch-I, and switch-II[35,56]. Through this engagement, the SAH-E acts as a regulatory "brake" to inhibit the ATPase activity of myosin VI by constraining the dynamic behavior of the active site. Additionally, given the importance of the L50 subdomain in modulating the actin-binding propensity of myosins[33,35,57], the close proximity of SAH-E to L50 may influence myosin VI's actin-binding, further contributing to a reduction in the actin-driven ATPase activity of the motor. Consistent with this unexpectedly structural finding, the mutations introduced in the SAH-E to disrupt its binding to the active site, dramatically elevated the ATPase activity of myosin VI (Fig. 3f). Collectively, our data demonstrate the indispensability of this helical sequence for the autoinhibition of myosin VI.

In addition to the direct impact of SAH-E on the MD, the intricate head/tail interaction in myosin VI may indirectly hinder ATPase activity by fixing the LA's orientation with the MD, as the coupling between ATP turnover and the LA's swing is a conserved feature of myosins[6,35]. By comparing the orientation of the LA in the autoinhibited state with those observed in the different states during the ATPase cycle of myosin VI[33,56–58] (Supplementary Fig. 9a, b), we found that, in the autoinhibited state, the LA's orientation closely resembles that of the PPS state (Supplementary Figs. 8a and 9g). However, as trapped in this orientation, Pi cannot be released and the LA cannot swing to adopt the ADP state (Supplementary Fig. 9g), resulting in the prolonged retention of hydrolysis products, ADP and Pi, at the active site and causing the ATPase cycle to stall. Once the autoinhibition of myosin VI

is lifted, the ATPase cycle can be promptly resumed with the release of Pi and ADP.

## The SAH and SAH-E constitute a regulatory module for myosin VI's autoinhibition and activation

In the autoinhibited state of myosin VI, the elongated SAH and SAH-E together act as a structural "spine" to integrate all individual domains into a compact myosin VI monomer (Fig. 3g). Conversely, the dynamic nature of SAH and SAH-E endows this "spine" to regulate the activation of myosin VI. For example, the SAH-E contains a ubiquitin-binding site[59] (Fig. 3g), in addition to another ubiquitin-binding site in CBD1[37]. It suggests that ubiquitin or ubiquitinated cargo may be involved in interfering with the head/tail interaction for the activation of myosin VI, aligning with the established role of myosin VI in autophagy[60,61]. Moreover, the binding of the SAH to LA requires the $Ca^{2+}$-free conformation of CaM2. Under condition of high $Ca^{2+}$ concentration, CaM2 may switch to the $Ca^{2+}$-bound conformation and thereby perturb the SAH/LA interaction (Fig. 3g), consistent with the role of $Ca^{2+}$ in activating myosin VI[31]. Additionally, the SAH stabilizes the 3HB structure of the LAE via hydrophobic interactions in the autoinhibited conformation of myosin VI (Fig. 2e). Given the propensity of LAE undergoing a compact-to-extended conformational transition upon encountering lipid environment[62], the recruitment of membranous cargos to myosin VI may disrupt the SAH/LAE interaction, serving as a trigger for myosin VI activation (Fig. 3g).

## Cargo-binding sites are sequestered in the autoinhibited state of myosin VI

More than a dozen of cargo/cargo adapters for myosin VI have been identified[14] (Fig. 1a). Cargo recognition in myosin VI involves two conserved motifs, RRL and WWY, located in CBD1 and CBD2, respectively[14] (Fig. 4a). Recent structural studies of myosin VI in complex with various cargo adapters have unveiled diverse modes of cargo recognition by CBD1 and CBD2[37–39,43,44] (Fig. 4a). By aligning the cargo-bound CBD structures with the autoinhibited myosin VI structure, we found that the cargo-binding sites in both CBDs are all blocked in the autoinhibited state (Fig. 4b–d). In CBD1, the SH3-like domain occupies one side of the RRL motif, while CBD2 occupies the other, effectively impeding the interaction with GIPC, CLCa, and ubiquitin (Fig. 4c). Similarly, in CBD2, the cargo-binding pocket shared by Dab2 and TOM1 is concealed by the CBD-linker (Fig. 4d). Additionally, another Dab2-binding site in CBD2 is partially obstructed by CaM2 in the LA (Fig. 4d). This extensive masking of cargo-binding sites in the autoinhibited state indicates that myosin VI autoinhibition may also hinder the CBD-dependent binding of other cargo adapters (e.g., OPTN, NDP52, TAX1BP1, and LMTK2)[63–66].

In line with our structural findings, autoinhibited myosin VI showed minimal interaction with GIPC, Dab2, and TOM1, as indicated by aSEC analysis in the PB condition (Fig. 4e). In contrast, myosin VI in the TB condition with the relatively open conformation (Supplementary Fig. 1b) readily formed complexes with both GIPC and Dab2 (Fig. 4f), presumably due to the alleviation of inhibitory head/tail interaction and CBD1/CBD2 stacking. Surprisingly, myosin VI in TB failed to interact with TOM1 (Fig. 4f), suggesting that the TOM1/Dab2-binding pocket in CBD2 remains blocked by the CBD-linker. Together, our observations demonstrate that the extensive intermolecular and interdomain interactions in the autoinhibited state of myosin VI effectively block the cargo-binding sites, preventing myosin VI from loading cargos unless prompted by activation cues.

## Cargo adapters release myosin VI autoinhibition by simultaneously disrupting multiple autoinhibitory interactions

Although cargo binding has been suggested as an important factor for releasing myosin VI autoinhibition[67–70], the underlying mechanism has remained unknown. Given that the binding of the cargo adapters to the CBD region structurally clashes with the autoinhibited conformation of myosin VI (Fig. 4c, d), we speculated that specific cargo binding may trigger the activation of myosin VI. To explore this possibility, we measured the ATPase activity of myosin VI in the presence of different cargo adapters. Notably, among the tested adapter proteins, GIPC and CLCa significantly increased the ATPase activity of myosin VI, while other adapters had more moderate effects (Fig. 4g). Our comparison of their binding modes revealed that GIPC has two distinct sites on CBD1, overlapping with the SH3-like/CBD1 and CBD1/CBD2 interfaces, respectively (Fig. 4c). It suggests that the binding of GIPC to CBD1 can disrupt both inhibitory interfaces. The binding of CLCa to CBD1 interferes with the SH3-like/CBD1 interface (Fig. 4c). However, due to the involvement of the large insertion (LI) region, the CLCa/CBD1 interaction results in the disruption of both the SH3-like/CBD1 and CBD1/CBD2 interfaces (Fig. 4c). Thus, the simultaneous disruption of multiple autoinhibitory interfaces mediated by cargo binding is likely a requisite for the efficient activation of myosin VI.

To further assess our hypothesis, we evaluated the activation effect of GIPC mutants (F319S and M276K) that individually disrupt its binding to either CBD1 surface (Fig. 4c). Indeed, each mutant alone failed to show the strong increase of the ATPase activity of myosin VI (Fig. 4h). Interestingly, the simultaneous addition of both mutant proteins restored the capability of GIPC to activate myosin VI (Fig. 4h), confirming the importance of the multiple interactions mediated by CBD1 in the transition between the autoinhibited and active states of myosin VI (Fig. 4i). Notably, despite the binding of Dab2 to CBD2 also interferes with two interfaces in the autoinhibited structure of myosin VI (Fig. 4d), the addition of Dab2 did not significantly increase myosin VI's ATPase activity, suggesting that cargo binding to CBD1, rather than CBD2, is crucial for the activation of myosin VI.

## Structural comparison of myosin II, V, and VI in the autoinhibited states

In addition to the structures of myosin VI reported here, the cryo-EM structures of myosin II and myosin Va, the other two prototype myosins representing conventional and unconventional myosins, respectively, in their autoinhibited state have been elucidated[26–30]. These myosins, despite having distinct tails, share commonalities in their autoinhibited conformations, such as the compact arrangement where the tail folds back onto the head, the formation of a hinge between the LA and tail, and the presence of a lengthy helical mediator connecting the tail and head (Fig. 5a). However, each myosin employs its unique autoinhibitory strategies, highlighting the diversity of regulatory mechanisms. A key difference lies in the orientations of their LAs (Fig. 5b). Although the MDs of both myosin II and myosin VI are trapped in the PPS state, their LAs extends in opposite directions, due to the unique In2 sequence in myosin VI that severs as a reverse gear for the walking direction[56]. Myosin Va, on the other hand, adopts a MD conformation resembling the Post-Rigor state, leading to an LA orientation distinct from those of myosin II or myosin VI. Furthermore, the hinge structures in these autoinhibited myosins are different (Fig. 5c). While myosin II and myosin Va directly employ light chain molecules for their hinge formation, and the two hinges are further stabilized through the following dimerized coiled-coil structure, myosin VI, in contrast, relies on its unique SAH and LAE for hinge stabilization in the monomeric structure. The head/tail interaction modes in the three myosins are also different (Fig. 5d). The N-terminal SH3-like domain is crucial for the head-to-tail interaction in myosin VI and myosin II, whereas myosin Va lacks such involvement of this domain. Additionally, the direct blockage of the active site in myosin VI is a unique feature absent in the other two myosins.

## Discussion

In summary, the high-resolution structure of myosin VI in its autoinhibited state presented in this work has unveiled the unprecedent

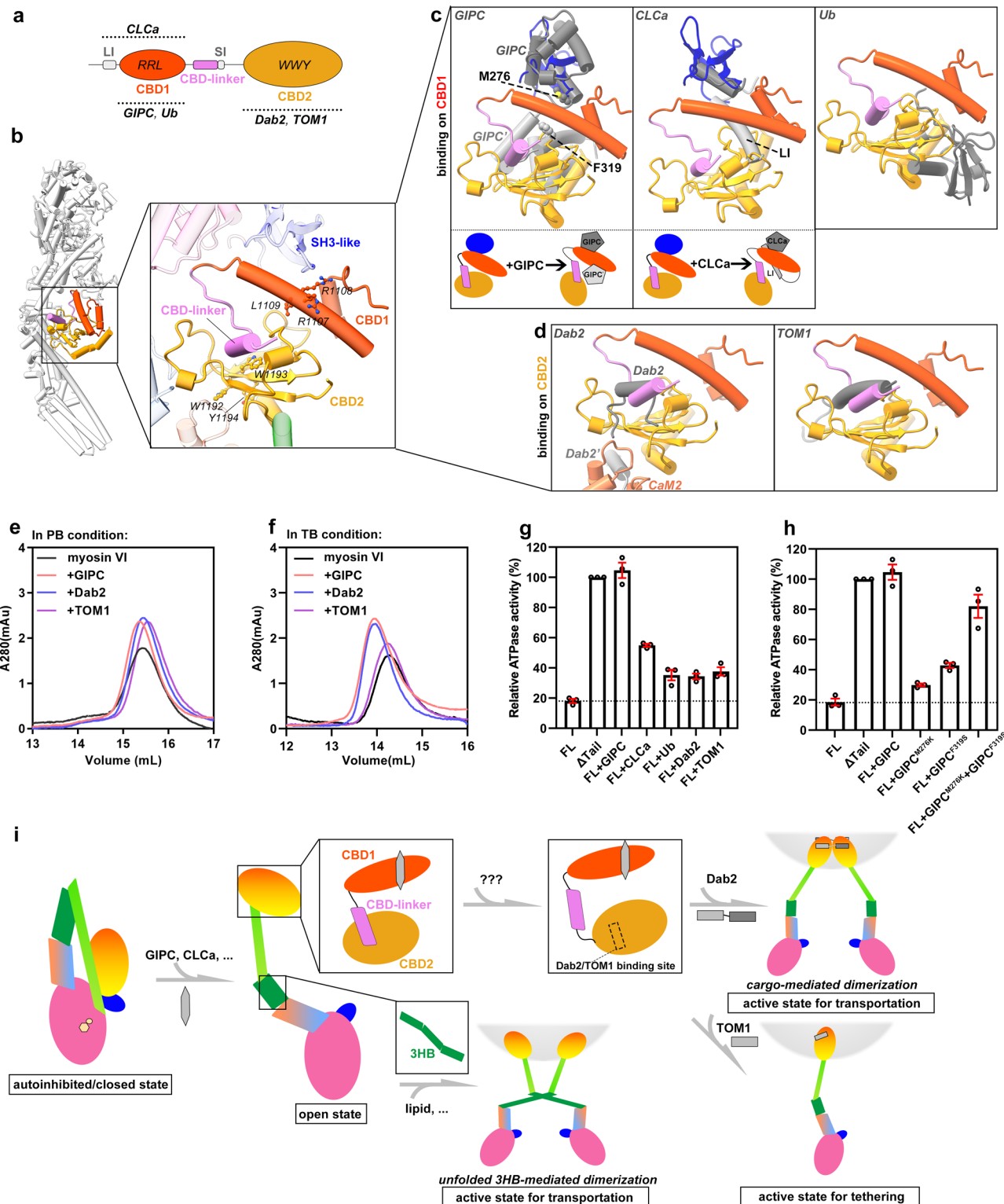

**Fig. 4 | Cargo-binding mediated disruption of autoinhibitory interfaces and myosin VI activation. a** Diagram of the CBD region showing the interactions of the five characterized cargos with CBD1 and CBD2 via the RRL and WWY motifs, respectively. **b** Two cargo-binding RRL and WWY motifs are blocked in the auto-inhibited myosin VI. **c, d** Structural alignments of CBD1 (**c**) and CBD2 (**d**) in the autoinhibited state with the corresponding cargo-bound CBDs. The overlapping between the autoinhibited structure and the bound cargo structures suggests the cargo-binding mediated activation of myosin VI. Cartoon diagrams show how GIPC or CLCa binding to CBD1 simultaneously disrupts two autoinhibitory interfaces. **e, f** aSEC analyses of the complex formation of myosin VI (1.5 μM) with three cargo adapters (15 μM) in TB (**e**) and PB (**f**). The Trx-tagged myosin VI-binding fragments of these adapter proteins were used for the analyses. **g** ATPase activities of myosin VI (0.2 μM) in the presence of indicated cargos (20 μM). **h** ATPase activities of myosin VI with GIPC or its mutants. In panels (**g**) and (**h**), the measured activity of the ΔTail mutant is set as 100% activation of myosin VI, and other samples are normalized to it for comparison. **i** Proposed model illustrating the transition from the autoinhibited to active states of myosin VI triggered by cargo binding.

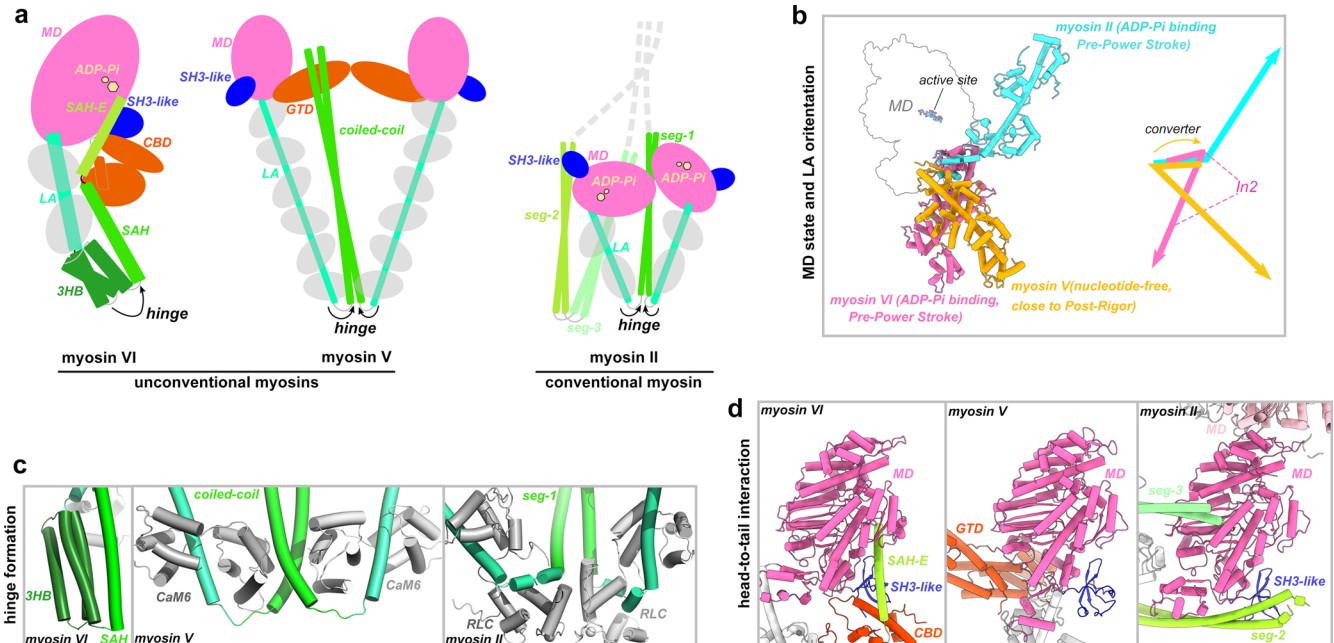

**Fig. 5 | Structural comparison of myosin VI, V, and II in the autoinhibited states. a** The schematic drawings of the three myosins in their autoinhibited conformations. The hinges are indicated. **b** Structural comparison of the MD structures and the corresponding LA positions in the autoinhibited structures. The different LA positions are illustrated by a schematic diagram. **c** Structural comparison of the different hinge formations in the autoinhibited structures. **d** Structural comparison of the head/tail interaction in myosin VI, myosin V, and myosin II, highlighting the differences. The models used for comparisons in panel **b**–**d** are the autoinhibited structures of myosin VI (PDB ID: 8W41), myosin Va (7YV9), and myosin II (7MF3).

insights into the intricate mechanisms governing the autoinhibition and activation of this unique motor protein. The distinct features revealed in the autoinhibited structure, including the crucial engagement of the SH3-like domain for the head-to-tail autoinhibition, the orchestrated obstruction of the motor activity with the elongated SAH and its extension, and the multifaceted assembly of the two CBDs through the CBD-linker, advance our understanding of the regulatory intricacies shared across diverse members of the myosin superfamily.

One of the major and unexpected discoveries in our study is the identification of the SAH-E and its direct interaction with the active site of the MD, adding a new layer of complexity to the autoinhibition of myosin VI. Despite its functional significance, the SAH-E/MD interaction exhibits a degree of flexibility, as indicated by a low percentage of particles in which the SAH-E sequence was observed in our EM analysis (Supplementary Fig. 6e). This flexibility is likely due, at least in part, to the dynamic nature of the SAH-E as a single helix, presumably enabling myosin VI to promptly respond to various cellular signals and environmental conditions, such as mechanical forces. We note with interests that other SAH-containing myosins, including myosin VII and X[51,52], have been suggested to adopt the monomeric "head-to-tail" architecture[23–25]. Thus, the essential role of SAH and SAH-E in coordinating the autoinhibited structure of myosin VI may provide a general understanding of the autoinhibition mechanism for this important subset of myosins.

In the cellular environment where ATP is abundant in the millimolar range, myosin VI predominantly assumes its autoinhibited state with ADP-Pi binding at active site, sequestering it away from inappropriate cargo, enabling rapid diffusion to sites where appropriate cargos need to be delivered, as well as preventing unnecessary ATP consumption. This regulatory mechanism empowers the cell to spatiotemporally control the activation of the motor protein. Our structural analysis strongly suggests that the loading of cargos to the CBDs triggers the activation of myosin VI. However, in our ATPase assay, only GIPC and CLCa can efficiently activate myosin VI (Fig. 4g), raising an intriguing question about how the diverse range of cargos activate

myosin VI. Considering the importance of the hinge formation and the potential role of lipids in unfolding the LAE[62], it is plausible that a membranous cargo containing multiple cargo adapters may simultaneously perturb several interfaces critical for the head/tail association and hinge structure and ensure the effective activation of myosin VI (Fig. 4i). Additionally, the phosphorylation of myosin VI may contribute to its activity control[71–73]. By mapping identified phosphorylation sites onto myosin VI, we found several sites are located at the autoinhibitory interfaces (Supplementary Fig. 12), suggesting a potential mechanism of myosin VI activation via phosphorylation modification of these sites.

Upon activation, monomeric myosin VI likely adopts an open conformation resembling the shape that we observed in the TB condition (Supplementary Fig. 1b). However, the processive walking of myosin VI for cargo transport along actin filaments requires its dimerization[74,75]. Two models were proposed to elucidate the transition of monomeric, activated myosin VI into its dimeric form. The first model suggests a cargo-mediated dimerization mediated by the binding of a cargo, such as Dab2, to CBD2 to form a 2:2 complex via a two-site binding mode[43,75,76]. However, presumably because the CBD-linker blocks the Dab2/TOM1-binding pocket, one of the Dab2-binding sites, on CBD2 even in the open conformation of myosin VI, Dab2-bound myosin VI predominantly remains the monomeric form (Fig. 4f). Thus, myosin VI dimerization likely involves the release of the CBD-linker from CBD2, mediated by an unknown mechanism (Fig. 4i). In addition, the release of the CBD-linker is also important for the loading of TOM1-mediated cargo to myosin VI for tethering function (Fig. 4i). The second model proposes a dimerization mechanism driven by the unfolded structure of 3HB in the LAE, which may be triggered by lipid binding[42,62] (Fig. 4i). Importantly, in both models, multiple stimuli including different cargos, cargo adapters, and other cellular environment are required for the cooperative and efficient activation of myosin VI for cargo transportation or tethering. The structural insights gained from our study provide crucial information for unveiling the intricate and precise nature of myosin VI regulation. In conjunction

with other autoinhibited structures of cytoskeletal motors[26–30], our results contribute to the broader understanding of these sophisticated nanomachines.

## Methods

### Plasmids

The gene encoding full-length myosin VI containing both the alternatively spliced large-insertion (LI) and small-insertion (SI) regions (GeneBank accession number NM_004999.4) was codon-optimized and synthesized by AZENTA. For protein expression and purification, myosin VI, the heavy chain, and its associated light chain, CaM, were cloned into a pCAG vector with a tandem His-Flag tag and a His tag at their N-termini, respectively. The constructs of myosin VI including the ΔTail (aa. 1–920), the MD (aa. 1–789), and the CBD1 (aa. 1072–1158) were cloned into pCAG vector and a modified pET32 vector, respectively. To investigate interactions with specific cargo adapters, the segments of mouse GIPC1 (aa. 258–333), mouse Dab2 (aa. 652–690), human TOM1 (aa. 437–463), human CLCa (aa. 46–61), and human ubiquitin (aa. 1–76) were constructed into a modified pET32 vector with a Trx-His tag at N-terminus. All the mutants were produced by the QuickChange Site-Directed Mutagenesis Kit, and all the plasmids were sequenced for verification.

### Protein expression and purification

Myosin VI or its mutants were co-transfected with CaM into the mammalian HEK293-F cells with the assistant of PEI transfection reagent. After ~60-h of transfection, cell pellets were collected and then manually lysed in an ice bath using a 40-mL tissue homogenizer in a lysis buffer containing 50 mM Tris (pH 7.5), 300 mM NaCl, 5 mM MgCl$_2$, 1 mM EGTA, 5 mM ATP, 2 mM DTT, 0.5% Triton™ X-100, and protease inhibitor cocktail. The lysate was centrifuged twice at 48,384 × $g$ for 1 h each and the supernatant was incubated with Flag-resins (GenScript) at 4 °C for 1 h. The resin-bound proteins were washed using a wash buffer containing 50 mM Tris (pH 7.5), 300 mM NaCl, 5 mM MgCl$_2$, 1 mM EGTA, 2 mM DTT and eluted in the wash buffer supplemented with 500 μg/mL Flag peptide at 4 °C. The eluted fractions containing the target protein were further purified using a size exclusion column (Superose 6 Increase 10/300 GL, GE) at 16 °C in either a phosphate buffer (PB) containing 20 mM Na$_2$HPO$_4$/NaH$_2$PO$_4$ (pH 7.4), 60 mM NaCl, 1 mM EGTA, 2 mM DTT, and 2 mM MgCl$_2$, or a tris buffer (TB) containing 50 mM Tris (pH 7.5), 80 mM NaCl, 1 mM EGTA, 2 mM DTT, and 2 mM MgCl$_2$. Finally, after confirmed by SDS-PAGE analysis, the fractions that contained the target protein were pooled and concentrated for subsequent cryo-EM and biochemical studies. To prepare the myosin VI$^{PB+ATP}$ sample, PB supplemented with 1 mM ATP was used throughout the whole purification process.

The CBD1 domain of myosin VI and the fragments of cargo adapters, including GIPC1, Dab2, TOM1, CLCa, and ubiquitin, which contain the myosin VI-binding sites, were expressed in *Escherichia coli* BL21(DE3) and purified by using Ni-NTA affinity chromatography followed by a size exclusion column (HiLoad 26/600 Superdex 200 pg, GE) in a buffer containing 50 mM Tris-HCl (pH 7.5), 100 mM NaCl, and 2 mM DTT at room temperature.

### Negative-staining EM analysis

The myosin VI sample in either TB or PB was applied onto a freshly glow-discharged grid with a carbon film at room temperature. After a 40-s incubation, excess protein solution was blotted away using a filter paper, and 2% uranyl acetate was then immediately applied to stain for one additional minute. After removing excess dye, the grid was air-dried at room temperature. All the prepared grids were checked, and ~50 micrographs were manually acquired for each sample using a Talos L120C G2 (FEI) operating at 120 kV. The collected micrographs were processed using CryoSPARC v4.0[77] with a negative-staining EM mode to conduct 2D classification.

### Cryo-EM grid preparation and data acquisition

All the protein samples were freshly prepared and 4 μL of each protein sample with the concentration of 0.4–0.5 mg/mL was applied onto a glow-discharged grid (QUANTIFOIL Cu, 300 mesh, 1.2/1.3). After a 5-s incubation, the grid was immediately blotted using filter paper for 3.5–4.5 s and quickly plunged into liquid ethane. All the processes were performed using a Vitrobot (FEI) under the condition of 4 °C (the measured pH value for TB is of 8.1) and 100% humidity. The prepared grids were stored in liquid nitrogen before screening and data collection.

Prior to data acquisition, the grids were transferred to a Titan Krios transmission electron microscope (Thermo Fisher Scientific) operating at 300 kV, equipped with a K3 direct electron detector. Movies for myosin VI$^{PB}$, myosin VI$^{TB+AMPPNP}$ and myosin VI$^{PB+ATP}$ were collected using SerialEM 3.7[78] in the counted super-resolution mode with a pixel size of 0.4135 Å. Each movie comprised of 32 frames was acquired in a 2.0-s exposure with a total dose rate of 50 e⁻/Å$^2$. The defocus range was set between −1.5 and −2.5 μm. Finally, 34196 movies of myosin VI$^{PB}$, 7138 movies of myosin VI$^{TB+AMPPNP}$ and 5801 movies of myosin VI$^{PB+ATP}$ were collected for cryo-EM structure determination. For the nucleotide-binding analysis, ~1000 movies for each sample in the indicated condition were collected for 2D and 3D classification analyses.

### Cryo-EM data processing

For each dataset of the samples including myosin VI$^{PB}$, myosin VI$^{TB+AMPPNP}$, and myosin VI$^{PB+ATP}$, similar processing procedures containing Motion correction, CTF estimation, particle picking, 2D classification, 3D classification, and the high-resolution 3D refinement were performed using cryoSPARC v4.0[77]. Briefly, movies were aligned using Patch motion correction with a B-factor of 150, resulting in micrographs with a physical pixel size of 0.827 Å. The defocus of each micrograph was estimated using CTFFIND4[79], and poor-quality micrographs were manually removed. Initial particle picking by Blob Picker results in 1,735,682 particles from 1688 micrographs of myosin VI$^{PB}$. The following 2D classification identified the open and the autoinhibited/closed states of myosin VI. With the open and the closed particles as the respective inputs, two deep-learning models were generated using Topaz Train[80] for particle picking. For all three datasets, the particles were picked using the two Topaz-trained models, and duplicates were removed with a minimum separation distance of 60 Å. The remaining particles were extracted and analyzed for 2D and 3D classifications. Especially, to optimize the orientation of the autoinhibited particles, after 2D classification, the particles with a fat shape at a specific view were selected to generate an additional Topaz-trained model. With particle picking guided by the Topaz model and the analysis of several rounds of 2D classification, the fat particles were selected to merge into the good classes of the particles from the previous 3D classification. After removing duplicates, high-resolution maps for myosin VI$^{PB}$, myosin VI$^{TB+AMPPNP}$, and myosin VI$^{PB+ATP}$ were generated using non-uniform refinement followed by local refinement in cryoSPARC. An additional local refinement focusing on the head region was applied to the dataset of myosin VI$^{PB+ATP}$ to obtain a 3.28-Å map of the head region with a density of ADP-Pi identified at the active site of the MD. In these processes, the box size of the extracted particles is set to be 440 × 440 pixels for the datasets of myosin VI$^{PB}$, myosin VI$^{TB+AMPPNP}$, and myosin VI$^{PB+ATP}$. To reduce the particle size and save the calculation source, we binned the particles by indicated times in the processes of 2D and 3D classifications.

Due to the identical conformation of myosin VI$^{PB}$, myosin VI$^{TB+AMPPNP}$, and myosin VI$^{PB+ATP}$, we combined the particles from the three datasets to further improve the density map of the autoinhibited structure. With one round of 2D and one round of 3D classifications in cryoSPARC, 272349 good particles were selected to load into Relion3.1[81]. On one hand, an additional 3D classification was applied with the settings of $T = 40$ and without alignment in Relion to further remove the poor particles. The best 3D class of 173658 particles was

selected and re-input into cryoSPARC to obtain the final 3.54-Å map of autoinhibited myosin VI. On the other hand, to filter out the particles containing the density of the SAH-E region, a focused 3D classification on the SAH-E was simultaneously performed in Relion with the settings of $T = 40$ and without alignment. 20386 particles containing the SAH-E density were selected to refine a map with the resolution of 4.14 Å in cryoSPARC. At last, all maps in this study were further optimized in the postprocess of DeepEMhancer[82] with a tightTarget model. The overall resolution for all the maps were calculated with the cut-off of FSC = 0.143 in cryoSPARC. The local resolution for the two final maps was calculated using the program of Local Resolution Estimation in cryoSPARC.

For the nucleotide-binding analysis, a same processing procedure containing Motion correction, CTF estimation, particle picking, 2D and 3D classifications was performed for each dataset of the indicated samples. Finally, a one round 3D classification with ten fixed initial models was applied to calculate the fraction of autoinhibited particles for each dataset.

### Model building and refinement
To build the atomic model of autoinhibited myosin VI, the previously-determined structures of the fragments including the head in the Pre-Power Stroke state (PDB: 2V26), the LA and LAE (PDB: 3GN4), the SAH (PDB: 6OBI), the CBD1 (PDB: 2N12), and the CBD2 (PDB: 2KIA) were fit into the final map at the 3.54-Å resolution in a mode of rigid-body to generate an initial model using UCSF-Chimera[83]. The initial model was manually adjusted in Coot[84], followed by real-space refinement in Phenix[85]. The CBD-linker was manually built based on the 3.54-Å density map. The SAH-E was assigned with the assistance of AlphaFold2-powered prediction[53,55] based on the 4.14-Å map. In addition, the atomic structure of ADP-Pi was added at the active site, as indicated by the map of myosin VI$^{PB+ATP}$. Finally, all these structural elements were integrated to produce the final model of the autoinhibited structure, which was further refined and validated against the 3.54-Å map.

The statistics information for cryo-EM data collection, processing, model refinement, and validation are summarized in Supplementary Table 1. All the figures displaying structural information were prepared by using UCSF ChimeraX[86].

### ATPase assay
The ATPase measurements were performed as previously reported[19]. In brief, the ATPase activity was measured at 25 °C in a buffer containing 4 mM MgCl$_2$, 0.25 mg/ml bovine serum albumin, 1 mM DTT, 2.5 mM phosphoenolpyruvate, 20 U/ml pyruvate kinase, 100 mM KCl, 1 mM EGTA, 10 µM actin, and 200 nM myosin VI proteins (wild-type or mutants) with either 20 mM Na$_2$HPO$_4$/NaH$_2$PO$_4$ (pH 7.4) or 50 mM Tris (pH 7.5). The reaction was initialized by adding 0.5 mM ATP and terminated by delivering 20 µL of reaction solution to a stop buffer containing 70 µL of 0.36 mM 2,4-dinitrophenyl hydrazine and 0.4 M HCl at specific time points of 5, 20, 40, 60, 80, 100, and 120 min. Following an additional incubation of the mixture at 37 °C for 10 min, 50 µL of color buffer containing 2.5 M NaOH and 0.1 M EDTA was added. The absorption at 460 nm was then measured using a Microplate reader (EnSpire). The reaction velocity was measured by linear regression of the A460 scores against reaction time. To standardize the results for comparison, ATPase activity was normalized to that of the fragmented myosin VI with deletion of the entire tail region that is considering to fully release autoinhibition and perform 100% activity. To quantify the impact of cargo adapters on the activation of myosin VI, each cargo adapter was added into the above system at a final concentration of 20 µM.

### Analytical size exclusion chromatography (aSEC)
The prepared samples with a volume of 100 µL in indicated conditions were loaded onto a Superose 6 Increase 10/300 GL or Superdex 200

Increase 10/300 GL column (GE Healthcare) on an ÄKTA pure system (GE Healthcare) at 16 °C, equilibrated with the PB (20 mM Na$_2$HPO$_4$/NaH$_2$PO$_4$, pH 7.4, 60 mM NaCl, 1 mM EGTA, 2 mM DTT, and 2 mM MgCl$_2$) or TB (50 mM Tris, pH 7.5, 80 mM NaCl, 1 mM EGTA, 2 mM DTT, and 2 mM MgCl$_2$) condition. To detect nucleotide-induced conformational change, myosin VI samples were pre-treated/supplied with a final concentration of 8 U/mL apyrase, 1 mM ADP/ATP, or 1 mM ATP analogs in TB. For cargo-binding assay, the myosin VI samples with a concentration of 1.5 µM were pre-incubated with 15 µM of the cargo proteins in the indicated TB or PB condition. For binding assay, 3.0 µM of the MD and CBD1 proteins were used in the indicated PB or TB conditions.

### Reporting summary
Further information on research design is available in the Nature Portfolio Reporting Summary linked to this article.

## Data availability
The cryo-EM density maps of autoinhibited myosin VI have been deposited into EMDB with accession codes EMD-37260 and EMD-37261. The corresponding atomic model has been also deposited in PDB with accession code 8W41. Source data for the figures and supplementary figures are provided as a Source Data file. Source data are provided with this paper.

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

## Acknowledgements

We thank Prof. Mingjie Zhang for his critical comments on our manuscript. We thank the assistance of Southern University of Science and Technology (SUSTech) Cryo-EM Centre and Core Research Facilities. Z.W. and C.Y. are investigators of SUSTech Institute for Biological Electron Microscopy. This work was supported by the National Natural Science Foundation of China (Grant No. 31971131 to Z.W. and 32170697 to C.Y.), Guangdong Basic and Applied Basic Research Foundation (Grant No. 2023A1515030232 to F.N.), Shenzhen Science and Technology Program (RCJC20210609104333007 to Z.W.), Shenzhen-Hong Kong Institute of Brain Science, Shenzhen Fundamental Research Institutions (2023SHIBS0002 to Z.W.), Shenzhen Key Laboratory of Biomolecular Assembling and Regulation (ZDSYS20220402111000001 to Z.W.), and Shenzhen Science and Technology Innovation Commission (JCYJ20200109141241950 to C.Y.).

## Author contributions

F.N., C.Y., and Z.W. conceived the study. C.Y. and Z.W. supervised the project. F.N., L.Li, L.W., J.X, S.X., Y.L., and L.Lin designed and performed experiments. F.N., L.Li, L.W., J.X, C.Y., and Z.W. analyzed the data. F.N., C.Y., and Z.W. wrote the manuscript with inputs from other authors.

## Competing interests

The authors declare no competing interests.
