## [Peer Review File · Nature Communications]

REVIEWER COMMENTS

Reviewer #1 (Remarks to the Author):

The noteworthy results of this paper are a first insight into the regulated (shutdown) structure of myoVI. This will be of interest to the myosin field, and those interested in the role of disease mutations in this myosin. Prior information about this structure was provided by much lower resolution negative stain EM, and thus this is a significance advance.

However, there are a few issues with the paper as outlined below:

The authors begin by describing the conditions under which auto-inhibited myosin VI forms. They note that more autoinhibited molecules form in PBS than in Tris-buffered solutions. Tris-buffered solutions are somewhat unusual to use for myosin as, the pH of Tris buffer is much more sensitive to temperature than PBS: a solution of pH 7.4 at room temperature increasing to pH 8 at 5 deg C, while the pH of PBS barely changes over the same temperature range. Is it more likely that myosin VI is sensitive to this pH change and changes conformation as a result? (temperature for experiments does not appear to be mentioned, other than that the grids were prepared at 4 deg C). Otherwise, it is hard to account for the effects of phosphate buffer vs Tris buffer.

The authors state that there is an identical conformation of MyoVIPB (is this Apo?) MyoVITB+AMPPNP, and MyoVIPB+ATP, which is a bit puzzling. Early negative stain EM of MyoVI (ref 21, and a follow up paper in Biophys J, Song et al., 2010) seem to show bent molecules in the Apo/ADP states, and extended molecules in ATP states. Likewise, the other reference (31) referred to by the authors also looks at myosin VI in the apo state and shows a bent conformation (all of their EM was performed in the absence of ATP), and not 'a straight shape' as cited here (line 88). While it might make sense for MyoVI to form an autoinhibited state in which ADP.Pi is in the nucleotide pocket, it's not clear to me that in the data presented here a) the resolution (at around 4 Angstroms) is high enough to be certain that this is the case. The data shown in Fig. 1b-d could be influenced by pH. Can the authors comment on this?

My recommendation would be to remove the Tris Data and just present the PB data. I don't believe that the Tris data adds anything significant to the paper and could be misleading.

Extended data fig 2 shows the general workflow for the CryoEM analysis, and Fig. 2d shows a small 'gold standard Fourier Shell Correlation plot. However, the authors omit to say if this is using CryoSPARC or Relion (or something else). What do they mean by 'gold standard?' CryoSPARC is generally thought to be somewhat more optimistic in estimating resolution. Can the authors specify? Similarly, Extended data fig

3 has a resolution (FSC) plot of 3.64Å – which software was used to calculate this? And again for Extended Data Figure 4. Elsewhere the authors do mention a lower resolution of around 4.1, which I suspect is more accurate.

Figure 1g is supposed to be an overlay of the map and the modelled structure, but it simply shows a rotated view of Fig. 1f. The contour levels for the EM density maps should be indicated in the legend for this and all the figures showing EM density maps.

The methods state that full length protein was expressed in mammalian cells. Do the authors know if some of the MyoVI is phosphorylated (or not?). (phosphorylation would activate MyoVI).

The statement (line 111) that 'In line with the critical role of γ -Pi binding in the autoinhibition of MyoVI, a distinct patch of density was observed at the canonical γ -Pi binding site within the active site of MyoVI in all three conditions (Extended Data Figs. 2c, 3c, and 4d), is just not supported by the data. The figure references also appear to be wrong, apart from 4d, which is so unclear (in terms of what density is what), it just can't be validated independently.

It would be helpful to clarify the methodology used to generate the crystal structure 4PK4 (extended data figure 6) by the Houdusse group. Effectively this structure was generated from a crystal of MyoVI in the apo state, after long soaking with PO₄, and it is not clear if this entirely recapitulates a PPS state, although there are changes around the nucleotide binding pocket.

The strongest argument for the 'pre-power' state argument is the position of the lever (extended data figure 7), and it would be helpful to mention this here, and definitely explain what position the converter domain and LAE is in, and how this corresponds to a pre-powerstroke state. Indeed, I failed to find any discussion on this, on reading through the paper.

Fig. 2 shows the various interfaces in the regulated myoVI structure and goes on to also look at ATPase measurements for different deafness mutations and the 'punctated' myoVI intensity (h) (g-i).

For clarity, I would separate out g-i into a separate figure, to enable Fig. 1 a-f to be made larger and clearer, and I'd avoid the use of yellow, which is hard to see.

I am not sure what 'punctated myosin VI intensity' means (Fig. 2h, Y-axis).

Lines 203 onwards: Is there any correlation between the change in ATP and the 'punctate' nature of GFP-MyoVI in cells? For example, W1115Q which increases ATPase activity, has no effect on 'punctate' localisation. Y1112C apparently decreases punctate localisation while increasing ATPase. How are these findings 'strongly aligned with the profound roles' of the head tail interaction? Even if the RRL interface is

compromised in these mutants, would you not expect them to be able to bind cargo and be 'punctate' through other CBD domains? The SAH-E mutations you make (Fig 3G) do result in more 'punctate' localisations, presumably because cargo binding is now allowed.

For the ATPase measurements, shown in Figs 1b and 2g, and elsewhere, I understand why you might want to normalise to the tail-less construct, but I think you should provide the ATPase rate for this construct. You should mention if it is similar to previous measurements. (e.g. by de la Cruz et al, JBC 2001). The increase in activity is only about 5 fold in PB buffer between FL and delta tail (Fig. 1b) – is this what you expect? Is it consistent with measurements by others? You need to state this.

The SAH-E interaction (Fig. 3D) is very interesting. But is SAH-E a misnomer? The charged region of myosin VI was originally identified in 2014 (Lister et al), and the term SAH was introduced to the field in 2015 (Knight et al., 2015) through work on the myosin X SAH domain. That paper shows an alignment between the SAH domains of VI, VII and X. SAH domains are rich in ERK residues (in fact the Spudich group renamed the myosin VI SAH as an 'ERK' helix in a later paper in PNAS). The SAH-E, while alpha helical is not as rich in E,R,K residues. Running the sequence through <https://waggawagga.motorprotein.de/> (which predicts SAH domains based on charge) suggests that residues beyond aa983 are below the cut off for a SAH. Thus SAH-E could be somewhat misleading. Why not just call it an alpha-helical extension?

Could you relate the orientation of the 2D class in Fig. 3a to the density map in Fig. 3b (seems to be rotated – good to know by how much).

Line 148: The statement: "Notably, while the SH3-like domain is present at the N-termini of many other myosins, its exact role in these myosins has not been elucidated". This is not correct, as the SH3 domain has been shown to play an important role in the shutdown state of smooth muscle myosin, and likely also in non-muscle myosin, through its interaction with the coiled coil.

However, later on (line 353) the comment is that the SH3 domain 'has a minor role in MyoII' Again, structure papers on autoinhibited MyoII do specifically mention that the coiled coil tail docks into a groove comprising partly of the SH3-like domain, and there are disease causing mutations in the SH3 like domain in non-muscle IIA.

In Fig. 5d: the comparison of the head-tail interactions: please state which structures were used for these. The structure for MyoII looks odd – as 'seg-2' seems to end on the SH3 like domain.

Line 382 The argument that the "abundance of ATP is why MyoVI predominantly

assumes its autoinhibited state, preventing needless ATP depletion without appropriate cargo” is unlikely to be correct. The levels of MyoVI in a cell are relatively low, and the amount of ATP they would consume, will also be really low. A single cell turns over about a billion molecules of ATP per second (Lemons et al., PLoS Biol, 2010). The numbers of myosin heads from MyoVI present in a cell, together with their combined rates of energy consumption is likely to be much less than 1% of this total turnover. The authors could perform this calculation to be convinced. It is much more likely that the regulated state is to sequester myosins away from cargo and enable rapid diffusion to where they are needed.

The paper ends rather abruptly, on dimerization by the LAE – which is relatively contentious. I suggest rewriting the end of the discussion to be more positive.

Minor:

Odd grammatical errors, English and odd phrases throughout, including the abstract, need to be fixed.

A unique “head-to-tail” architecture (Fig. 1e, f), in which MyoVI ties its “head” with the “tail” through the interaction between the very N-terminal SH3-like domain and CBD1.’ ‘very N-terminal’ makes no sense!

Temperatures of experiments should be provided throughout.

Nikorn A1R confocal microscope with 100x objective lens

I think this should be Nikon, and the NA of the lens should be stated

Reviewer #2 (Remarks to the Author):

The authors describe in this study the cryoEM structure of the autoinhibited MYO6, which gives insight into the molecular detail on the interaction between the motor and tail domain. They identify the regions in the tail that inhibit ATPase activity and show that cargo-binding sites in tail domain are blocked in the folded conformation. The authors further demonstrate that GIPC and clathrin light chain binding

leads to release of the autoinhibited conformation, which at least for GIPC has been shown previously. Despite this, the experimental results are novel and provide an important contribution to our understanding of the regulation of motor function in the cellular environment.

I am not able to comment on the quality and execution of the cryo-EM experiments, as I am an expert in this technique. The description of the structural data and the interactions between the different MYO6 domains, however, is very well done, described in detail and easy to visualise and to understand using the structures provided.

The finding that the N-terminal SH3-domain interacts with the cargo-binding domain in the cryo structure is interesting. Do these two domains interact when incubated in isolation in vitro, which would be an elegant way to verify the head-tail interaction?

The title is slightly misleading as the cryo-EM structure shows the autoinhibited state of MYO6, however, the process of activation can only be assumed from the accessibility of the cargo adaptor binding site.

Line 57-59: There are no cellular investigations in these references to suggest that the tail folds back on the MYO6 motor domain.

Is there any explanation why MYO6 adopts different conformations in Tris or Phosphate buffer?

Analysis of disease-causing MYO6 mutations measuring ATPase activity is sufficient in vitro, however, the analysis of these MYO6 mutations in cells assessing the “puncta formation” requires further analysis. The authors should demonstrate that these “puncta” are vesicular structures, performing double labelling experiments with endosomal marker proteins. This is required to exclude that the MYO6 mutants form punctate aggregates in cells. Furthermore, in order to quantitate the number of puncta shown in Fig. 2g, the expression levels of the different constructs have to be normalised.

In figure 4 h, please explain why the GIPC double mutant, which no longer binds to MYO6, causes a stronger activation of MYO6 than the single mutants. The GIPC double mutant that no longer binds to either CBD1 interface causes activation of MYO6 similar to the wildtype GIPC? This requires explanation.

Figure 5 does not contain any experimental data, and thus is better suitable as a figure in a review article.

Minor points:

The English language requires correction in places, for example “With the physiological ATP level, autoinhibited MyoVI adopts a compact, monomeric conformation ...”

“proper cargos”

The nomenclature should for human myosin VI should be used correctly according to HGNC, the approved human gene nomenclature. For the human protein used in this study (NM_004999.4) it should be myosin VI or MYO6. The nomenclature used by the authors, MyoVI, is highly unusual.

The labelling for the residues maintaining the autoinhibited state and those in that are a deafness mutation are very difficult to differentiate in Extended data Fig. 8.

Figure 2g, the explanation for box around mutations is missing.

Labelling of panels in figure 5 should be in small letters.

Reviewer #3 (Remarks to the Author):

The manuscript by Niu et al describes the high resolution cryoEM structure of the auto-inhibited conformation of myosin VI. This structure is very significant since it is the first high resolution structure of an auto-inhibited “monomer” myosin motor. The other two members of the myosin superfamily for which there are similar resolution structures are either from Class II or Class V motors which are both dimeric with respect to their motor domains. There are several interesting features of the structure that distinguish the autoinhibitory mechanism from those other myosins. First, a unique role for the SH3 moiety at the N-terminus is described. The mechanism for stabilizing the hairpin turn is unique as is a role for the SAH-E in stabilizing the inhibition of the motor. The last two features would not be expected to be found in myosins II or V since these structural elements are missing. Other nice features are the

resolution of the Pi in the active site and the complementation of the structural results with in vitro and in vivo measurements of function of mutations. The paper is timely and will be of great interest to a wide array of biochemists, structural biologists and cell biologists. I do have several concerns and comments that should be addressed.

Several bits of information that one usually sees in a structural paper are missing. It would be good to see a figure showing the resolution at the various domains of the molecule and a movie showing a rotation of the molecule along its long axis would be useful to the reader. I assume that the resolution is not uniform in all bits of the structure and it would be nice to know how much confidence one can apply to the various statements that have been made.

Also, the authors show and comment on specific side chains in many of the figures, but it is not clear that these are supported by clear densities. It would be useful to see density maps for some of these. If the resolution is not sufficient to clearly define side chain positions, and if the proposed interactions are based on molecular dynamic modeling or some other tool, please state.

In many cases the figure legends are not sufficiently detailed to understand all the nuances of figures. Specific examples will be given below.

I could not find where the aSEC experiments were described in the Methods. This should be provided. Specifically, I would like to see the buffer conditions for TB and PB. Were ATP included in these buffers?

I'm also curious as to how ionic strength affects the "open" and "closed" states as determined using the aSEC method. Does the molecule extend in high ionic strength even when ATP is added?

It would be of interest to see the density assigned to the phosphate in your structures. How does the position of this phosphate compare to that of the Houdusse structure that was soaked in phosphate?

For the Figures containing ATPase values, give the value for the 100% sample normalization.

It would be interesting to know how well the auto-inhibited myosin VI binds to actin in the presence of ATP, given that its actin binding regions are largely exposed. In the myosin II auto-inhibited structures the cleft between the upper and low 50 kdomain of the free head remains open as seen in the prepowerstroke crystal structures which likely explains its weak affinity for actin. Could you specifically comment on the state of the cleft in myosin VI?

The experiments in Fig. 4 with binding partners were carried out at a single (albeit rather high) concentration of binding partners. Is it possible that the reason for some of the lower activation values could be incomplete saturation of the myosin VI by the binding partner?

Also, according to the literature if any of the binding partners dimerize myosin VI, then you would expect only 50% or the activity of the delta tail construct. Has this been considered?

Figure 2g,h: I figured this out without too much trouble, but it would have been quicker if the figure legend had told me that the “i-1, i-2” notions in the X-axis legend corresponded to the interfaces described in the figures in panels a-g.

Fig. 4i. What are the “starburst-like” symbols representing in some of the panels/

Minor Points:

Line 100: I presume this experiment was carried out in Tris Buffer. If so, please insert “in Tris Buffer” in between “particles increased”.

Line 371. I assume that you mean “a low percentage of particles in which the SAH-E sequence is observed”. Presumably, all particles contain the sequence, but it not resolved in most of them.

Point-by-Point Response

Before our response to the comments, we thank all three reviewers for their recognition of the novelty and significance of our works and their critical and constructive suggestions that help us to efficiently improve the science of the manuscript. During revision, we have performed additional experiments and analyses to address questions raised by the reviewers. Due to the data update, all figures and their panels mentioned in our response follow the new arrangement in the revised manuscript, unless otherwise indicated.

Reviewer #1 (Remarks to the Author):

The noteworthy results of this paper are a first insight into the regulated (shutdown) structure of myoVI. This will be of interest to the myosin field, and those interested in the role of disease mutations in this myosin. Prior information about this structure was provided by much lower resolution negative stain EM, and thus this is a significance advance.

However, there are a few issues with the paper as outlined below:

The authors begin by describing the conditions under which auto-inhibited myosin VI forms. They note that more autoinhibited molecules form in PBS than in Tris-buffered solutions. Tris-buffered solutions are somewhat unusual to use for myosin as, the pH of Tris buffer is much more sensitive to temperature than PBS: a solution of pH 7.4 at room temperature increasing to pH 8 at 5 deg C, while the pH of PBS barely changes over the same temperature range. Is it more likely that myosin VI is sensitive to this pH change and changes conformation as a result? (temperature for experiments does not appear to be mentioned, other than that the grids were prepared at 4 deg C). Otherwise, it is hard to account for the effects of phosphate buffer vs Tris buffer.

Response: The reviewer pointed out the possible influence of pH changes in the Tris-buffer condition at different temperatures. However, the observed conformational change of myosin VI is unlikely caused by the use of Tris-buffer. As indicated in Fig. 1c, myosin VI shows obvious conformational change upon the addition of ATP analogs. These experiments were performed under identical TB conditions at the same temperature of 16°C. Moreover, both negative staining (prepared with TB at room temperature, see Extended Data Fig.1b) and cryo-EM (prepared with TB at 4°C, see Fig. 1d) samples of myosin VI in TB condition showed the “bent-shaped” conformation, which is consistent with the previous observation of the Apo state using MOPS or Tris buffer (*Song CF, et al. Biophys J. 2010 Nov 17;99(10):3336-44; Batters C, et al. Proc Natl Acad Sci U S A. 2016 Mar 1;113(9):E1162-9.*). Altogether, temperature-induced pH change in TB is unlikely to drive the conformational transition between open and closed states. Noteworthy, as indicated by our biochemical and EM analysis (Fig. 1c and 1d), the binding of phosphate (in the PB condition) or phosphoryl group (in ATP or ATP analogs) to the active site is likely the primary factor to trigger the conformational change.

To enhance clarity, we have emphasized the role of phosphate binding for the conformational

transition in the Result, supplemented the Methods section with detailed temperature information for all experiments, and enriched the experimental procedures for the aSEC analysis. Furthermore, we have added the labels for the TB condition in Fig.1c and revised the figure legend as “The samples in TB were prepared either without additive or with an additional 8U/mL apyrase, 1 mM ADP/ATP, or 1 mM ATP analogs. The samples in PB were prepared by exchanging buffer from TB to PB.”.

The authors state that there is an identical conformation of MyoVI^{PB} (is this Apo?) MyoVI^{TB+AMPPNP}, and MyoVI^{PB+ATP}, which is a bit puzzling. Early negative stain EM of MyoVI (ref 21, and a follow up paper in Biophys J, Song et al., 2010) seem to show bent molecules in the Apo/ADP states, and extended molecules in ATP states. Likewise, the other reference (31) referred to by the authors also looks at myosin VI in the apo state and shows a bent conformation (all of their EM was performed in the absence of ATP), and not ‘a straight shape’ as cited here (line 88). While it might make sense for MyoVI to form an autoinhibited state in which ADP.Pi is in the nucleotide pocket, it’s not clear to me that in the data presented here a) the resolution (at around 4 Angstroms) is high enough to be certain that this is the case. The data shown in Fig. 1b-d could be influenced by pH. Can the authors comment on this?

Response: We acknowledge the potential confusion about our description of myosin VI^{PB} in the original manuscript. Here, despite the absence of ATP or its analogs, myosin VI^{PB} adopts the conformation resembling the ATP state rather than the Apo state, based on our negative stain EM (Extended Data Fig.1c) and cryo-EM (Extended Data Fig.2) analyses. To further display the consistency in the motor domain between myosin VI^{PB} and the ATP state in myosin VI^{TB+AMPPNP} and myosin VI^{PB+ATP}, we have included an additionally local map comparison focusing on the active site of the motor domain. This comparison, as shown in Extended Data Fig. 5a, highlights the overlap of the bound Pi in myosin VI^{PB} with the position of γ -Pi of the bound ADP-Pi in myosin VI^{PB+ATP}. This observation confirms that the binding of Pi to the active site drives the motor domain into the Pre-Power stroke (PPS) state. In line with our observation, the crystal structure of the myosin VI motor domain bound with soaked Pi adopts the PPS state (*Llinas P, et al. Dev Cell. 2015 May 26;33(4):401-12.*). In this structure, the Pi molecule occupies the identical position to what we observed in the PB condition (Extended Data Figs.5a and 7a). Given the requirement of the PPS state in the autoinhibited conformation (Extended Data Fig. 8c), the presence of Pi in the PB condition contributes to the formation of the autoinhibition or ATP state of myosin VI by promoting the PPS state of the MD. To emphasize this point, we have added more detailed explanation in the first two sections of the Results part.

In addition, as suggested by Reviewer 2, we have investigated the head/tail interaction by using the purified fragments of the motor domain and the CBD1. As shown in Extended Data Fig. 8d and 8e, we found that the PB, but not TB condition, promotes the MD/CBD1 interaction. This further suggests that the autoinhibition formation of myosin VI requires the PPS state of the MD, facilitated by the binding of Pi to the active site in the typical binding position of γ -Pi.

We modeled ADP-Pi in the myosin VI^{PB+ATP} structure for two main reasons. Firstly, compared to ATP, ADP-Pi offers a better fit to the observed density in the nucleotide-binding pocket (see Fig. R1 below). Secondly, previous studies have suggested that the MD in the PPS state binds

to the ATP hydrolysis products, ADP and Pi, as shown in the crystal structure where ADP-VO₄³⁻ was used as a well-accepted mimic of ADP-Pi (Ménétreay, J., et al. *Cell*. 2007, 131, 300-8).

*Fig. R1. Density-fit-model to represent ADP-Pi and ATP at the active site.
(map counter level=0.1).*

My recommendation would be to remove the Tris Data and just present the PB data. I don't believe that the Tris data adds anything significant to the paper and could be misleading.

Response: We appreciate the reviewer's concern regarding the experiments performed in the TB condition. However, referring to our response to the first question above, we have demonstrated that the conformational transition of myosin VI is triggered by the binding of γ -Pi and is not an artifact induced by the TB condition. Noteworthy, the myosin VI sample in TB shows the apo state (nucleotide-free) conformation (Extended Data Fig.1b). It serves as an essential control, allowing us to analyze the conformational transition induced by the addition of ATP and its analogs (Fig. 1c and 1d), as well as the cargo binding capability (Fig. 4g and 4f). Omitting this part of data would pose difficulties in explaining the conformational dynamics of myosin VI, thereby significantly compromising the clarity and coherence of our manuscript. Given these considerations, we decided to retain the Tris data.

To enhance clarity and avoid any potential misinterpretation of the Tris data, we revised our description of the experiments performed in TB and added more explanations, as mentioned in our responses to the first two concerns above.

Extended data fig 2 shows the general workflow for the CryoEM analysis, and Fig. 2d shows a small 'gold standard Fourier Shell Correlation plot. However, the authors omit to say if this is using CryoSPARC or Relion (or something else). What do they mean by 'gold standard?' CryoSPARC is generally thought to be somewhat more optimistic in estimating resolution. Can the authors specify? Similarly, Extended data fig 3 has a resolution (FSC) plot of 3.64Å – which software was used to calculate this? And again, for Extended Data Figure 4. Elsewhere the authors do mention a lower resolution of around 4.1, which I suspect is more accurate.

Response: In our cryo-EM data processing, except for the focused 3D classifications without alignment showed in Extended Data Fig. 5c and 5f, all procedures, including the final map refinement and map resolution estimation, were performed using cryoSPARC software. Follow the review's suggestion, we have explicitly indicated the use of cryoSPARC v4.0 in the procedures of 3D refinements (Extended Data Fig. 5c and 5f). Furthermore, we clarified in the Methods with the sentences: "The best 3D class of 173,658 particles was selected and re-input into cryoSPARC to obtain the final 3.54-Å map of autoinhibited myosin VI" and "20,386 particles containing the SAH-E density were selected to refine a map with a resolution of 4.14

Å in cryoSPARC”. Additionally, we have mentioned “The local resolution for the two final maps was calculated using the program Local Resolution Estimation in cryoSPARC” in the Methods.

The term “gold standard” here refers to the resolution at $FSC = 0.143$, a standard commonly used in cryoSPARC as GSFSC resolution, presented in Extended Data Figs. 2d, 3d, 4e, 5e and 5h. To eliminate any potential confusion caused by the lack of axis labels in these figures, we have now indicated the y-axis as Fourier Shell Correlation and the x-axis as Resolution. Moreover, we have explicitly stated that the cut-off resolution at $FSC = 0.143$ in these figures, and this information has been added to the Methods with the sentence: “The overall resolution for all the maps were calculated with the cut-off of $FSC = 0.143$ in cryoSPARC”.

To avoid the possibility of optimistic resolution calculation in cryoSPARC, we also refined the two final maps (with/without SAH-E density) using the same particle sets and masks in Relion software. As shown in Fig. R2 below, the two maps were generated with overall resolution of 3.64 and 4.18 Å at the cut-off of $FSC = 0.143$, which are comparable to the corresponding map resolutions of 3.54 and 4.14 Å obtained in cryoSPARC, respectively. This consistency confirms the reliability of the map resolution calculations in our study. As suggested, we have stated in the Methods that cryoSPARC was used for all map resolution calculations.

Fig. R2. Two maps with overall resolution of myosin VI generated in Relion3.1 software. (map counter level = 0.1)

Figure 1g is supposed to be an overlay of the map and the modelled structure, but it simply shows a rotated view of Fig. 1f. The contour levels for the EM density maps should be indicated in the legend for this and all the figures showing EM density maps.

Response: Indeed, a rotation has been applied between Fig. 1g and 1f to optimize the presentation for both the overall structure and ADP-Pi. In the revised Fig. 1g, we have labeled the rotation angles to provide clarity on this presentation. In addition, as suggested, we have indicated the contour levels for all represented density maps in the corresponding legends of our revised manuscript.

The methods state that full length protein was expressed in mammalian cells. Do the authors know if some of the MyoVI is phosphorylated (or not?). (phosphorylation would activate MyoVI).

Response: The reviewer raised an interesting point regarding phosphorylation regulation of myosin VI. Previous reports also indicated that phosphorylation may contribute to the activation of myosin VI in cells. Thus, during revision, we mapped the identified phosphorylation sites derived from PhosphoSitePlus database (www.phosphosite.org, references ≥ 2) onto myosin VI (see Fig. R3a below). Among these sites, S1019, Y1112, Y1137, and Y1182 were found to be located at the autoinhibitory interfaces (see Fig. R3b), supporting the potential role of phosphorylation modification in regulating myosin VI's activation. We have included this part of analysis as Extended Data Fig. 11 and discussed the potential regulation of myosin VI activation via phosphorylation in the Discussion.

Fig. R3. Potential phosphorylation sites in myosin VI

The statement (line 111) that ‘In line with the critical role of γ -Pi binding in the autoinhibition of MyoVI, a distinct patch of density was observed at the canonical γ -Pi binding site within the active site of MyoVI in all three conditions (Extended Data Figs. 2c, 3c, and 4d), is just not supported by the data. The figure references also appear to be wrong, apart from 4d, which is so unclear (in terms of what density is what), it just can’t be validated independently.

Response: We thank the reviewer for pointing out this issue. To present our observations at the active sites more clearly, we have included additional figures to show the local maps of myosin

VI in all three conditions with the focus on this region (Extended Data Fig. 5a). The local maps show the ambiguous density of the bound Pi locating at the γ -Pi positions of AMPPNP, and ADP-Pi molecules. Also, the structural elements at the active site in all three conditions adopt the essentially same conformation upon Pi or γ -Pi binding. To accurately reflect our findings, we have accordingly modified the statement and the citing figure: “In line with the critical role of γ -Pi binding in the autoinhibition of myosin VI, a similar patch of density was observed at the canonical γ -Pi binding site within the active site of myosin VI in all three conditions (Extended Data Fig. 5a)”.

It would be helpful to clarify the methodology used to generate the crystal structure 4PK4 (extended data figure 6) by the Houdusse group. Effectively this structure was generated from a crystal of MyoVI in the apo state, after long soaking with PO₄, and it is not clear if this entirely recapitulates a PPS state, although there are changes around the nucleotide binding pocket.

Response: To address the concern raised, we have included the ADP.VO₄-bound motor structure (PDB ID: 2V26) in our structural comparison (Extended Data Fig. 7a) in the revised manuscript. This addition ensures a more thorough representation of the conformational similarity of the MD in the autoinhibited and PPS states. In addition, we have clarified in the revised figure legend of Extended Data Fig. 7a that the motor conformation observed in 4PK4 results from soaking in a condition containing Pi.

The strongest argument for the ‘pre-power’ state argument is the position of the lever (extended data figure 7), and it would be helpful to mention this here, and definitely explain what position the converter domain and LAE is in, and how this corresponds to a pre-powerstroke state. Indeed, I failed to find any discussion on this, on reading through the paper.

Response: We thank the reviewer for the wonderful suggestion. To better address this argument, we have compared the MD structures in previously-identified states and the autoinhibited state in the revised manuscript as Extended Data Fig. 8c. This structural comparison shows that only in the PPS state, the converter in the MD is rotated outward, preventing clashes with the tail region in autoinhibited myosin VI while allowing the MD/CBD interaction. We have also provided additional context in the “The N-terminal SH3-like domain mediates the head/tail interaction (interface-1)” section of the revised manuscript to further address the significance of the PPS state in the autoinhibited state formation of myosin VI.

Fig. 2 shows the various interfaces in the regulated myoVI structure and goes on to also look at ATPase measurements for different deafness mutations and the ‘punctated’ myoVI intensity (h) (g-i). For clarity, I would separate out g-i into a separate figure, to enable Fig. 1 a-f to be made larger and clearer, and I’d avoid the use of yellow, which is hard to see.

Response: As suggested, we have enlarged the panels a-f in the revised Fig. 2 and replaced the yellow color with orange throughout all the related figures, including Figs 1a, 1e, 1f, 2b, 2c, 2g, 2h, 3g, 4a, 4b, 4i, and Extended Data Figs. 5i and 9.

I am not sure what ‘punctated myosin VI intensity’ means (Fig. 2h, Y-axis).

Lines 203 onwards: Is there any correlation between the change in ATP and the ‘punctate’

nature of GFP-MyoVI in cells? For example, W1115Q which increases ATPase activity, has no effect on ‘punctate’ localization. Y1112C apparently decreases punctate localization while increasing ATPase. How are these findings ‘strongly aligned with the profound roles’ of the head tail interaction? Even if the RRL interface is compromised in these mutants, would you not expect them to be able to bind cargo and be ‘punctate’ through other CBD domains? The SAH-E mutations you make (Fig 3G) do result in more ‘punctate’ localizations, presumably because cargo binding is now allowed.

Response: We thank the reviewer for pointing out the potential controversy in our interpretation of the cellular data, which was also raised by Reviewer 2. Our original thought was that activated myosin VI could bind to vesicular cargos and thereby accumulate on vesicles to form puncta, which however has not been validated. Thus, to explore the relationship between myosin VI puncta and endosomes, as suggested by Reviewer 2, we have conducted a cellular analysis using the endosomal marker Rab5. Upon overexpressing both the active mutant of myosin VI and Rab5, we found that the I929Q mutant, carrying a mutation in the SAH region without impacting cargo binding, forms puncta in the cytosol. However, although these myosin VI puncta can colocalize with Rab5 puncta in some HeLa cells, most transfected cells showed little colocalization of these two puncta (see Fig. R4 below), suggesting that the punctate distribution of myosin VI may not be simply linked to protein accumulation on vesicles. To avoid potential controversy, we have removed our cellular results, including the original Figs. 2h, 3g and Extended Data Fig. 10, from our revised manuscript. Considering the clear and supportive evidence from the biochemical assays, we believe that the removal of this part of data will not significantly weaken our structural findings about the autoinhibition mechanism of myosin VI.

Fig. R4. HeLa cells co-expressing GFP-myosin VI-I929Q and DsRed-Rab5.

For the ATPase measurements, shown in Figs 1b and 2g, and elsewhere, I understand why you might want to normalise to the tail-less construct, but I think you should provide the ATPase rate for this construct. You should mention if it is similar to previous measurements. (e.g. by de la Cruz et al, JBC 2001). The increase in activity is only about 5 fold in PB buffer between FL and delta tail (Fig. 1b) – is this what you expect? Is it consistent with measurements by others? You need to state this.

Response: The primary aim of our ATPase assay in this study is to efficiently and directly

validate the autoinhibitory interfaces observed in our cryo-EM structure. For this purpose, we performed a comparative analysis of the ATPase activities between wild-type and many mutants under the condition with a fixed concentration of F-actin (10 μ M). Although we didn't calculate the exact ATPase rate for each construct, it allows us to effectively analyze the relative effects of disrupting different autoinhibitory interfaces. Our ATPase analysis aligns with the well-established method of PEP-PK coupled ATP regeneration system, which has also been widely used in myosin field (*Li XD, et al. Biochem Biophys Res Commun. 2004;315(3):538-545.; Umeki N, et al. Nat Struct Mol Biol. 2011;18(7):783-788.; Lu Z. et al. J Biol Chem. 2014 Jun 27;289(26):18535-48.*). Compared with the assay used by de la Cruz *et al*, our assay has a slight modification in the coloring method (NADH system used in the JBC 2001 paper, whereas we used 2,4-dinitrophenyl hydrazine to estimate the product pyruvate). To provide more clarity, we have included the corresponding citation for the ATPase assay in the Methods.

As suggested, we have searched literature and found that approximately 7-fold and 5-fold increases in ATPase activity have been detected between the full-length protein and the tail-less fragment for myosin V and myosin X, respectively (*Li XD, et al. Biochem Biophys Res Commun. 2004;315(3):538-545.; Umeki N, et al. Nat Struct Mol Biol. 2011;18(7):783-788.*), which are comparable to the 5-fold increase in this study. We have stated in the revised manuscript that our measurement of ATPase activity is comparable with previous studies.

The SAH-E interaction (Fig. 3D) is very interesting. But is SAH-E a misnomer? The charged region of myosin VI was originally identified in 2014 (Lister *et al*), and the term SAH was introduced to the field in 2015 (Knight *et al.*, 2015) through work on the myosin X SAH domain. That paper shows an alignment between the SAH domains of VI, VII and X. SAH domains are rich in ERK residues (in fact the Spudich group renamed the myosin VI SAH as an 'ERK' helix in a later paper in PNAS). The SAH-E, while alpha helical is not as rich in E,R,K residues. Running the sequence through <https://waggawagga.motorprotein.de/> (which predicts SAH domains based on charge) suggests that residues beyond aa983 are below the cut off for a SAH. Thus SAH-E could be somewhat misleading. Why not just call it an alpha-helical extension?
Response: We thank the reviewer's recognition of our SAH-E finding. Regarding the nomenclature, we agree that the SAH-E contains no typical SAH sequence pattern. However, we chose this name to highlight its direct extension from the SAH domain in myosin VI, similar to the naming of the lever arm extension (LAE) for the 3HB region. Meanwhile, we have clarified in the revised manuscript that "We henceforth termed this helix as SAH-extension (SAH-E), which lacks a typical SAH sequence pattern rich in E, R, and K residues".

Could you relate the orientation of the 2D class in Fig. 3a to the density map in Fig. 3b (seems to be rotated – good to know by how much).

Response: Indeed, it has a rotation of 90 degrees between the 2D class in panel a and the density map in panel b in the original version. To enhance the clarity, in the revised Figure 3, we have added a density map with an identical orientation observed in the 2D class and indicated the rotation angle between the new density map and the original one to aid readers in understanding the results.

Line 148: The statement: “Notably, while the SH3-like domain is present at the N-termini of many other myosins, its exact role in these myosins has not been elucidated”. This is not correct, as the SH3 domain has been shown to play an important role in the shutdown state of smooth muscle myosin, and likely also in non-muscle myosin, through its interaction with the coiled coil.

However, later on (line 353) the comment is that the SH3 domain ‘has a minor role in MyoII’ Again, structure papers on autoinhibited MyoII do specifically mention that the coiled coil tail docks into a groove comprising partly of the SH3-like domain, and there are disease causing mutations in the SH3 like domain in non-muscle IIA.

Response: We thank the reviewer for pointing out our incorrect descriptions about the SH3-like domain. We have accordingly revised the sentences as “Notably, the SH3-like domain in the conventional myosin II is important for the autoinhibition formation, while the role of the SH3-like domain in many unconventional myosins has not been elucidated.” and “The N-terminal SH3-like domain is crucial for the head-to-tail interaction in myosin VI and myosin II, whereas myosin Va lacks such involvement of this domain”.

In Fig. 5d: the comparison of the head-tail interactions: please state which structures were used for these. The structure for MyoII looks odd – as ‘seg-2’ seems to end on the SH3 like domain.

Response: As suggested, we have optimized the orientation of the myosin II structure in Fig.5d to avoid any potential confusion. In addition, we have added the PDB IDs of the myosin VI, V, and II structures in the figure legend.

Line 382 The argument that the “abundance of ATP is why MyoVI predominantly assumes its autoinhibited state, preventing needless ATP depletion without appropriate cargo” is unlikely to be correct. The levels of MyoVI in a cell are relatively low, and the amount of ATP they would consume, will also be really low. A single cell turns over about a billion molecules of ATP per second (Lemons et al., PLoS Biol, 2010). The numbers of myosin heads from MyoVI present in a cell, together with their combined rates of energy consumption is likely to be much less than 1% of this total turnover. The authors could perform this calculation to be convinced. It is much more likely that the regulated state is to sequester myosins away from cargo and enable rapid diffusion to where they are needed.

Response: We agree with the reviewer that the autoinhibition state may save only a small portion of the whole ATP pool. However, the autoinhibition of motors may still offer a way to avoid energy waste in cells, particularly in certain conditions with insufficient energy supply. Thus, as suggested, we have modified the argument as “In the cellular environment where ATP is abundant in the millimolar range, myosin VI predominantly assumes its autoinhibited state, sequestering it away from inappropriate cargo, enabling rapid diffusion to sites where appropriate cargos need to be delivered, as well as preventing unnecessary ATP consumption”. In addition, we have also revised the sentence “To prevent unnecessary ATP consumption when proper cargos are absent, many myosins ...” to “In the absence of cargos, many myosins ...” in the Introduction.

The paper ends rather abruptly, on dimerization by the LAE – which is relatively contentious. I suggest rewriting the end of the discussion to be more positive.

Response: Following the reviewer's suggestion, we have included further discussion here to emphasize the complexity of activity regulation and our contribution to advancing the understanding of myosin regulation.

Minor:

Odd grammatical errors, English and odd phrases throughout, including the abstract, need to be fixed.

Response: As suggested, we have carefully gone throughout the manuscript to improve the language.

A unique "head-to-tail" architecture (Fig. 1e, f), in which MyoVI ties its "head" with the "tail" through the interaction between the very N-terminal SH3-like domain and CBD1.' 'very N-terminal' makes no sense!

Response: We have deleted the word of "very" in this sentence.

Temperatures of experiments should be provided throughout.

Response: We have added the information of experiment temperatures in the Methods.

Nikorn A1R confocal microscope with 100x objective lens

I think this should be Nikon, and the NA of the lens should be stated

Response: As mentioned above, we have removed the cellular results from our revised manuscript, and accordingly, the corresponding description in the Methods has been deleted.

Reviewer #2 (Remarks to the Author):

The authors describe in this study the cryoEM structure of the autoinhibited MYO6, which gives insight into the molecular detail on the interaction between the motor and tail domain. They identify the regions in the tail that inhibit ATPase activity and show that cargo-binding sites in tail domain are blocked in the folded conformation. The authors further demonstrate that GIPC and clathrin light chain binding leads to release of the autoinhibited conformation, which at least for GIPC has been shown previously. Despite this, the experimental results are novel and provide an important contribution to our understanding of the regulation of motor function in the cellular environment.

I am not able to comment on the quality and execution of the cryo-EM experiments, as I am an expert in this technique. The description of the structural data and the interactions between the different MYO6 domains, however, is very well done, described in detail and easy to visualize and to understand using the structures provided.

The finding that the N-terminal SH3-domain interacts with the cargo-binding domain in the cryo structure is interesting. Do these two domains interact when incubated in isolation in vitro, which would be an elegant way to verify the head-tail interaction?

Response: What a wonderful suggestion! As the suboptimal protein quality of the SH3-like fragment for aSEC analysis, we used the MD containing the SH3-like domain to explore its interaction with the CBD1 fragment. As expected, we observed an obvious binding between the MD and the CBD1 in the PB condition (Fig. R5a). Remarkably, this binding was disrupted by the interface mutation W1115Q in CBD1, as revealed by our autoinhibited structure (Fig. R5b), further confirming our structural observation of the head-tail interaction. Furthermore, we found that this interaction did not occur in the TB condition (Fig. R5c). Since Pi binding to the active site in PB promotes the Pre-Power stroke (PPS) state formation of the MD of myosin VI, this result further supports the notion that the head-tail interaction requires the PPS state of the MD. By comparing the MD structures in different states, we found that the converter of MD in rigor, post-rigor, and ADP-binding states would clash with the tail region, providing a rationale for the requirement of the PPS state of the MD in the autoinhibition formation of myosin VI. These findings have been incorporated into the revised manuscript as Extended Data Fig. 8c-f and detailed in the “The N-terminal SH3-like domain mediates the head/tail interaction (interface-1)” section of the Results.

Fig. R5. aSEC analysis of the MD and CBD1 interaction in PB and TB conditions.

The title is slightly misleading as the cryo-EM structure shows the autoinhibited state of MYO6, however, the process of activation can only be assumed from the accessibility of the cargo adaptor binding site.

Response: We respectfully appreciate the reviewer's comment on the title issue. Our rationale for the chosen title is from the critical insights provided by our cryo-EM structure, into the activation process of myosin VI. Firstly, as pointed out by the reviewer, the autoinhibited structure unveils blocked cargo adaptors binding sites (Fig. 4b), offering an activation mechanism through cargo binding. Combined with our biochemical characterization (Fig. 4e-h), the structural analysis of disruptive impact of cargo binding on the autoinhibited state (Fig. 4c and 4d) indicates that cargo adaptors can release myosin VI autoinhibition by simultaneously disrupting multiple autoinhibitory interactions. Secondly, our structure uncovers the importance role of hinge formation in maintaining the autoinhibition of myosin VI (Fig. 2e). Given the potential influence of lipids in unfolding the LAE (*Mukherjea, M. et al. Mol Cell, 2009, 35, 305-15; Yu, C. et al. J Biol Chem, 2012, 287, 35021-3503*), our structural findings propose a plausible mechanism that a membranous cargo containing cargo adaptors may perturb several interfaces critical for the head/tail association and hinge structure, ensuring the effective activation of myosin VI. Thirdly, our structure unexpectedly reveals the vital role of SAH-E in myosin VI autoinhibition (Fig. 3). As the SAH-E contains a ubiquitin binding site (*Penengo, L. et al. Cell. 2006, 124, 1183-1195*), this finding suggests the involvement of ubiquitinated cargos in activating myosin VI, aligning with the established role of myosin VI in autophagy. Taken together, the autoinhibited structure of myosin VI provides key information, significantly advancing our understanding of both the autoinhibition and activation mechanisms.

Line 57-59: There are no cellular investigations in these references to suggest that the tail folds back on the MYO6 motor domain.

Response: We apologize for the incorrect statement. We have removed the words "and cellular" from the sentence.

Is there any explanation why MYO6 adopts different conformations in Tris or Phosphate buffer?

Response: In our cryo-EM map of myosin VI in the PB condition, a phosphate density occupies the typical γ -Pi position of bound ATP at the active site (Extended Data Figs. 2d and 5a). As ATP addition can induce the pre-power stroke (PPS) state of the MD (*Lister I, et al. EMBO J. 2004 Apr 21;23(8):1729-38*), it is plausible that the binding of phosphate to the active site also promotes the formation of the PPS state. Our structural analysis reveals the necessity of the PPS state of MD for the autoinhibition formation of myosin VI (Extended Data Figs. 7a and 8c). Thus, the abundant phosphate in the PB condition binding to the active site drives the autoinhibition formation of myosin VI. However, in TB condition, the MD without phosphate binding prefers the rigor state, which interferes with the head-tail interaction (Extended Data Fig. 8d-f), leading to the open state of myosin VI. This explanation has been added into the revised manuscript.

Analysis of disease-causing MYO6 mutations measuring ATPase activity is sufficient in vitro, however, the analysis of these MYO6 mutations in cells assessing the "puncta formation"

requires further analysis. The authors should demonstrate that these “puncta” are vesicular structures, performing double labelling experiments with endosomal marker proteins. This is required to exclude that the MYO6 mutants form punctate aggregates in cells. Furthermore, in order to quantitate the number of puncta shown in Fig. 2g, the expression levels of the different constructs have to be normalized.

Response: We thank the reviewer for pointing out the potential controversy in our interpretation of the cellular data, which was also raised by Reviewer 1. Our original thought was that activated myosin VI could bind to vesicular cargos and thereby accumulate on vesicles to form puncta, which however has not been validated. Thus, to explore the relationship between myosin VI puncta and endosomes, as suggested, we have conducted a cellular analysis using the endosomal marker Rab5. Upon overexpressing both the active mutant of myosin VI and Rab5, we found that the I929Q mutant, carrying a mutation in the SAH region without impacting cargo binding, forms puncta in the cytosol. However, although these myosin VI puncta can colocalize with Rab5 puncta in some HeLa cells, most transfected cells showed little colocalization of these two puncta (see Fig. R6 below), suggesting that the punctate distribution of myosin VI may not be simply linked to protein accumulation on vesicles. To avoid potential controversy, we have removed our cellular results, including the original Figs. 2h, 3g and Extended Data Fig. 10, from our revised manuscript. Considering the clear and supportive evidence from the biochemical assays, we believe that the removal of this part of data will not significantly weaken our structural findings about the autoinhibition mechanism of myosin VI.

Fig. R6. HeLa cells co-expressing GFP-myosin VI-I929Q and DsRed-Rab5.

In figure 4 h, please explain why the GIPC double mutant, which no longer binds to MYO6, causes a stronger activation of MYO6 than the single mutants. The GIPC double mutant that no longer binds to either CBD1 interface causes activation of MYO6 similar to the wildtype GIPC? This requires explanation.

Response: We acknowledge the potential misleading label in Figure 4h. The accurate representation is that we used a mixture of two mutant GIPC proteins, M276K and F319S, which individually lose the ability to interact with myosin VI on one interface while retaining binding on the other. This complementary mixture mimics the wild-type GIPC, effectively disrupting both interfaces between CBD1 and the SH3-like domain and between CBD1 and CBD2. Therefore, the addition of this mixture results in a substantial increase in ATPase activity

of myosin VI, comparable to the wild-type GIPC, confirming that multi-interface disruptions are essential for myosin VI activation. To avoid any confusion, we have updated the label with “FL+GIPC^{M276K}+GIPC^{F319S}” in figure 4h.

Figure 5 does not contain any experimental data, and thus is better suitable as a figure in a review article.

Response: Figure 5 provides a novel comparison of unconventional myosin VI, myosin V and conventional myosin II, offering detailed structural insights into their similarities and variations in forming autoinhibition states. As commented by Reviewer 3-“This structure is very significant since it is the first high resolution structure of an auto-inhibited ‘monomer’ myosin motor. The other two members of the myosin superfamily for which there are similar resolution structures are either from Class II or Class V motors which are both dimeric with respect to their motor domains.”, this comparative analysis facilitates to expand our understanding of the myosin superfamily. Given its importance and relevance to interested readers, we decide to retain Figure 5 in our manuscript.

Minor points:

The English language requires correction in places, for example “With the physiological ATP level, autoinhibited MyoVI adopts a compact, monomeric conformation ...”, “proper cargos”

Response: As suggested, we have carefully gone throughout the manuscript to improve the language.

The nomenclature should for human myosin VI should be used correctly according to HGNC, the approved human gene nomenclature. For the human protein used in this study (NM_004999.4) it should be myosin VI or MYO6. The nomenclature used by the authors, MyoVI, is highly unusual.

Response: As suggested, we have replaced “MyoVI” with “myosin VI” throughout the revised manuscript.

The labelling for the residues maintaining the autoinhibited state and those in that are a deafness mutation are very difficult to differentiate in Extended data Fig. 8.

Response: We have enhanced the visibility of these disease mutations by using red arrowheads marked with the mutation name to indicate their positions.

Figure 2g, the explanation for box around mutations is missing.

Response: In the original version, we mentioned the information about the box around these deafness mutations in the Figure 2h legend. To further clarify it, we have added this information in the legends of panel 2g.

Labelling of panels in figure 5 should be in small letters.

Response: We have corrected it in the revised Fig. 5.

Reviewer #3 (Remarks to the Author):

The manuscript by Niu et al describes the high resolution cryoEM structure of the auto-inhibited conformation of myosin VI. This structure is very significant since it is the first high resolution structure of an auto-inhibited “monomer” myosin motor. The other two members of the myosin superfamily for which there are similar resolution structures are either from Class II or Class V motors which are both dimeric with respect to their motor domains. There are several interesting features of the structure that distinguish the autoinhibitory mechanism from those other myosins. First, a unique role for the SH3 moiety at the N-terminus is described. The mechanism for stabilizing the hairpin turn is unique as is a role for the SAH-E in stabilizing the inhibition of the motor. The last two features would not be expected to be found in myosins II or V since these structural elements are missing. Other nice features are the resolution of the Pi in the active site and the complementation of the structural results with in vitro and in vivo measurements of function of mutations. The paper is timely and will be of great interest to a wide array of biochemists, structural biologists and cell biologists. I do have several concerns and comments that should be addressed.

Several bits of information that one usually sees in a structural paper are missing. It would be good to see a figure showing the resolution at the various domains of the molecule and a movie showing a rotation of the molecule along its long axis would be useful to the reader. I assume that the resolution is not uniform in all bits of the structure and it would be nice to know how much confidence one can apply to the various statements that have been made.

Response: We thank the reviewer for pointing out the issue about the missing information in our original manuscript. The resolution maps for our autoinhibited myosin VI have been presented in Extended Data Fig. 5d and 5g, showing the local resolution for each domain of myosin VI. Additionally, to enhance clarity and visual understanding, we have included two supplementary movies (Extended Data Movie 1 and 2) in our revised manuscript, illustrating the two cryo-EM maps and corresponding overall structures for autoinhibited myosin VI with and without the SAH-E, respectively.

Also, the authors show and comment on specific side chains in many of the figures, but it is not clear that these are supported by clear densities. It would be useful to see density maps for some of these. If the resolution is not sufficient to clearly define side chain positions, and if the proposed interactions are based on molecular dynamic modeling or some other tool, please state.

Response: In addition to the density presentation for CBD-linker region in Extended Data Fig. 5i, we have included the density maps for key structural elements in a model-fit-density pattern to validate and represent our structure quality (Extended Data Fig.6).

In many cases the figure legends are not sufficiently detailed to understand all the nuances of figures. Specific examples will be given below.

I could not find where the aSEC experiments were described in the Methods. This should be provided. Specifically, I would like to see the buffer conditions for TB and PB. Were ATP included in these buffers?

Response: As suggested, we have included additional details on the aSEC assay in the Methods.

It's crucial to note that neither TB nor PB condition in this assay contains ATP. Also, we have carefully checked and optimized the figure legends to ensure they provide sufficient information.

I'm also curious as to how ionic strength affects the "open" and "closed" states as determined using the aSEC method. Does the molecule extend in high ionic strength even when ATP is added?

Response: Following the reviewer's suggestion, we have tested the potential effect of ionic strength on the transition between the "open" and "closed" state using aSEC. As shown in Fig. R6 below, with an increase in NaCl concentration, myosin VI tends to transit from a closed to an open state. However, compared to the open state as indicated by myosin VI in the TB condition, even 500 mM NaCl cannot completely disrupt the closed state (Fig. R7), suggesting that ATP binding plays a more pronounced influence than ionic strength in promoting the autoinhibition of myosin VI.

Fig. R7. aSEC profiles of myosin VI in different salt and buffer conditions.

It would be of interest to see the density assigned to the phosphate in your structures. How does the position of this phosphate compare to that of the Houdusse structure that was soaked in phosphate?

Response: Following the reviewer's suggestion, we have performed a comparison between our autoinhibited myosin VI structure and the Houdusse structure (PDB ID: 4PK4). We found a positional overlap between the phosphate in our structure and the soaked phosphate (Extended Data Fig. 7a). Interestingly, both phosphates occupy the same position as the VO_4^{3-} ion in the crystal structure of the MD with the Pre-Power stroke (PPS) state (PDB ID: 2V26) (Extended Data Fig. 7a). The consistent observation of the PPS state in these structures suggests that the presence of phosphate at the active site can promote the MD to adopt the PPS state.

For the Figures containing ATPase values, give the value for the 100% sample normalization.

Response: In the ATPase results presented in Figs. 2g, 3f, 4g and 4h, the ATPase values were normalized to the activity of the tail-deletion construct (ΔTail), presumed to represent the active

sate, set as 100% ATPase activity for comparisons. This information has been clarified in the legends of the respective figures with the statement: “The measured activity of the Δ Tail mutant is set as 100 % activation of myosin VI and other samples are normalized to it for comparison”.

It would be interesting to know how well the auto-inhibited myosin VI binds to actin in the presence of ATP, given that its actin binding regions are largely exposed. In the myosin II auto-inhibited structures the cleft between the upper and low 50 k domain of the free head remains open as seen in the prepowerstroke crystal structures which likely explains its weak affinity for actin. Could you specifically comment on the state of the cleft in myosin VI?

Response: To explore the actin binding ability of the autoinhibited myosin VI, we have compared our structure with the crystal structure of the MD in the PPS state. As shown in Extended Data Fig 7a, the conformation of the actin-binding site in these structures is highly similar. Since the PPS state is known in the myosin field to have a weak binding ability to actin filaments (*Robert-Paganin, J., et al. Chem Rev. 2020, 120, 5-35*), it is likely that autoinhibited myosin VI also have a weak affinity for actin. In the revised manuscript, we have accordingly added the comment “Meanwhile, considering the PPS state has been demonstrated in myosin field to have a weak binding ability to actin filament, the PPS state of the MD observed in our structure suggests that myosin VI in its autoinhibited conformation may weakly associate with actin.” into the Results.

The experiments in Fig. 4 with binding partners were carried out at a single (albeit rather high) concentration of binding partners. Is it possible that the reason for some of the lower activation values could be incomplete saturation of the myosin VI by the binding partner?

Response: In this ATPase assay, the cargo concentration (20 μ M) is 100-fold higher than that of myosin VI (0.2 μ M). It is very unlikely that the lower activation values result from incomplete saturation of myosin VI by the binding partner. Given that the binding affinities to the CBD regions of myosin VI are in the sub- μ M range for not only GIPC (*Rai et al. J Biol Chem. 2022;298:101688.*) but also Dab2 (*Yu et al. Cell. 2009;138:537-548.*) and TOM1 (*Hu et al. Nat Commun. 2019;10:3459.*), the high cargo over myosin ratio ensures the saturated binding of cargos to activated myosin VI.

Also, according to the literature if any of the binding partners dimerize myosin VI, then you would expect only 50% or the activity of the delta tail construct. Has this been considered?

Response: Indeed, the “gating” effect of dimerized myosin motors is a crucial consideration in ATPase measurements. However, as shown in Fig. 4e and 4f, myosin VI does not dimerize in the presence of different cargos, despite Dab2 has been proposed to mediate myosin VI dimerization. As we mentioned in the manuscript, “presumably because that the CBD-linker blocks the Dab2/TOM1-binding pocket, one of the Dab2 binding sites, on CBD2 even in the open conformation of myosin VI, Dab2-bound myosin VI predominantly remains the monomeric form (Fig. 4f)”. Thus, we proposed that “myosin VI dimerization likely involves the release of the CBD-linker from CBD2, mediated by an unknown mechanism (Fig. 4i)”. Since no dimer formation was observed in our assays, we did not account for the “gating” effect in our ATPase measurements.

Figure 2g,h: I figured this out without too much trouble, but it would have been quicker if the figure legend had told me that the “i-1, i-2” notions in the X-axis legend corresponded to the interfaces described in the figures in panels a-g.

Response: Following the reviewer’s suggestion, we have added the description of “The labels ‘i-1’ to ‘i-6’ in panels **g** and **h** correspond to the six essential inhibitory interfaces shown in panels **a** to **f**.” in the revised figure legend.

Fig. 4i. What are the “starburst-like” symbols representing in some of the panels/

Response: These “starburst-like” symbols were initially used to indicate the disruptive effects on the interfaces of the autoinhibited structure upon myosin VI activation. Since these symbols may lead to misinterpretation and the conformational changes drawn in this model may adequately convey the interface disassembly, we have deleted these “starburst-like” symbols in the revised Figure 4.

Minor Points:

Line 100: I presume this experiment was carried out in Tris Buffer. If so, please insert “in Tris Buffer” in between “particles increased”.

Response: As suggested, we have inserted “in the TB condition” in this sentence.

Line 371. I assume that you mean “a low percentage of particles in which the SAH-E sequence is observed”. Presumably, all particles contain the sequence, but it not resolved in most of them.

Response: We have updated this sentence as suggested.

REVIEWER COMMENTS

Reviewer #1 (Remarks to the Author):

The paper is much improved from its original version. It was a good decision to remove the cell biology, which detracted from the overall paper.

I just have a few minor comments:

The question about pH has not been answered well: I think the authors have misunderstood my original comment. To reiterate:

They say that more autoinhibited molecules form in PBS than in Tris-buffered solutions. In Tris buffered solution, the pH could be 8 at 5 deg C, but in PBS it will still be around 7. The higher number of autoinhibited molecules in PBS, could therefore arise as a result (partly) of a difference in pH. The authors should state in the main text what the pH is at the temperature used, in the two different buffers as evidence that it is not pH, as they claim, but more likely to be occupation of the active site with phosphate, if this is indeed the case.

The density for the phosphate (or AMPPNP or ADP.Pi) is supposedly demonstrated in extended figure 5A. However, this figure has three density maps superimposed on each other, with 'zoom-in views with the phosphates and nucleotides overlapped by their density' to the side. What we need to see is the density overlapping the structure in these zoom-in views, otherwise there is no evidence for what's being shown (is there actual density for AMPPNP etc). I think this is important, so it needs to be shown clearly. (Fig. 1d also does not show this). However, in the cell, I assume it is more likely that ADP.Pi is in the nucleotide binding site, and this should be mentioned. Similarly, the ADP.Pi is 'shown' in extended Fig. 4d – but again -this seems to be as part of the workflow for generating the structure – and it's impossible to see from this where/if ADP is actually present.

I recommend a stand-alone extended figure that clearly shows the density and structures, for Pi, AMPPNP and ADP.Pi, with overlap and zoom in to definitely show that there is density for the specific nucleotides.

Moreover, I also asked about the phosphorylation state of Myosin VI. I did not mean by this to ask if there are phosphorylation sites in MyoVI (as they have answered, and as indeed has already been published) but if they have looked to see if the MyoVI they purified is phosphorylated? However, they

have not done this. Do they know? Again, I think this is important to know and relatively simple to find out.

In extended figure 8a, an ATPase cycle is shown with arrows, that seems to suggest the cycle progresses from rigor to post rigor to prepower stroke to ADP binding. I'm afraid I don't quite understand this, because the pre-power stroke should be ATP/ADP.Pi (so why not have ATP/ADP.Pi here) and the next step is not 'ADP binding' (as the ADP doesn't bind), rather Pi is lost leaving ADP behind. Can the authors modify this diagram to make it clearer. Moreover, in the myosin field, we normally consider the cycle as starting with the pre-power stroke, not rigor. The LA wouldn't swing back to the ADP state, as the lever position in the ADP state is after phosphate release (line 275) – so the description here seems to be a bit confusing. This logic just needs making a bit clearer, and the diagram also made clearer.

Line 363: The statement 'Although the MDs of both myosin II and myosin VI are trapped in the PPS state, their LAs extends in opposite directions, due to the unique In2 sequence in myosin VI' seems a bit odd – as the unique sequence is exactly what makes Myosin VI a 'backwards motor'. So, clearly the lever is going to be differently oriented in the PPS. Perhaps at line 40, a short sentence that explains that the insert positions the lever in such a way as to drive its movement to towards the pointed end of F-actin in contrast to other myosins, would be useful background information, with a short follow up here. (Helpful for people not directly in the myosin field).

I also don't understand the statement 'While myosin II and myosin Va employ light chain molecules to shape their hinges, and the coiled coil dimerizes hinges for further stabilization'. This seems wrong to me: the light chains are not bound to a hinge, and I'm not sure what 'dimerised' hinges means or why this might stabilise the molecule. The key difference here is that myosin V and II are dimeric, and myosin VI isn't! This statement needs removing, or changing.

Perhaps also of note here is that kinesin-1 does use a mechanism to block its active site in its shutdown state (e.g. work by Hackney and others)

Just a small comment: The use of the term 'lever arm' is incorrect, and 'lever' should be used instead: A lever is a rigid body capable of rotating on a point on itself. A lever arm is the perpendicular distance from the fulcrum of a lever to the line of action of the effort and (in physics) includes the idea of the amount of force needed to generate (or prevent) movement of an object around an axis. I realise that lever arm is generally used in the myosin field, but it's actually wrong – and should just be lever.

While I'm happy that the authors have tempered their language for the autoinhibited state to enable diffusion, rather than conserve ATP, it would have been good if they had actually done the calculation – just how much energy does it save to have Myosin VI in an autoinhibited state?

Reviewer #2 (Remarks to the Author):

The authors have taken my suggestions and comments on board and have revised the manuscript accordingly.

Reviewer #3 (Remarks to the Author):

The authors have addressed the issues I raised and I have no further questions.

Point-by-Point Response

Reviewer #1 (Remarks to the Author):

The paper is much improved from its original version. It was a good decision to remove the cell biology, which detracted from the overall paper.

I just have a few minor comments:

The question about pH has not been answered well: I think the authors have misunderstood my original comment. To reiterate:

They say that more autoinhibited molecules form in PBS than in Tris-buffered solutions. In Tris buffered solution, the pH could be 8 at 5 deg C, but in PBS it will still be around 7. The higher number of autoinhibited molecules in PBS, could therefore arise as a result (partly) of a difference in pH. The authors should state in the main text what the pH is at the temperature used, in the two different buffers as evidence that it is not pH, as they claim, but more likely to be occupation of the active site with phosphate, if this is indeed the case.

Response: The reviewer's request is to highlight the potential pH effect on the significant difference in the autoinhibited state between PB and Tris buffer in the main text. However, the difference has already been rigorously established at room temperature, with both buffers having the same pH value. This is supported by various analyses, including analytical gel-filtration (Extended Data Fig. 1a), negative staining EM (Extended Data Fig. 1b, c), and ATPase activity assay (Fig. 1b). Therefore, emphasizing the pH issue in the main text that is unrelated to our main findings may not be warranted. Nevertheless, we have included the measured pH information (7.6 and 8.1) for Tris buffer at both room temperature (for negative staining sample preparation) and 4°C (for cryo-EM sample preparation) in the Method, allowing readers to assess potential pH effect themselves.

The density for the phosphate (or AMPPNP or ADP.Pi) is supposedly demonstrated in extended figure 5A. However, this figure has three density maps superimposed on each other, with 'zoom-in views with the phosphates and nucleotides overlapped by their density' to the side. What we need to see is the density overlapping the structure in these zoom-in views, otherwise there is no evidence for what's being shown (is there actual density for AMPPNP etc). I think this is important, so it needs to be shown clearly. (Fig. 1d also does not show this). However, in the cell, I assume it is more likely that ADP.Pi is in the nucleotide binding site, and this should be mentioned. Similarly, the ADP.Pi is 'shown' in extended Fig. 4d – but again -this seems to be as part of the workflow for generating the structure – and it's impossible to see from this where/if ADP is actually present.

I recommend a stand-alone extended figure that clearly shows the density and structures, for PI, AMPPNP and ADP.Pi, with overlap and zoom in to definitely show that there is density for the specific nucleotides.

Response: Following the reviewer's suggestion, we have included a stand-alone extended figure (Extended Data Fig. 5) to illustrate the comparison of the three maps. Panel a shows the overlapping of the maps, panel b presents each individual map in a map-fit-model mode, and

panel c provides zoom-in views of the active site in a density-fit-model mode, indicating the similarity of three overall maps and the different densities for the bound phosphate and nucleotides at the active sites. In addition, we have adjusted the related description in the Discussion, stating that “In the cellular environment where ATP is abundant in the millimolar range, myosin VI predominantly assumes its autoinhibited state with ADP-Pi binding at active site”.

Moreover, I also asked about the phosphorylation state of Myosin VI. I did not mean by this to ask if there are phosphorylation sites in MyoVI (as they have answered, and as indeed has already been published) but if they have looked to see if the MyoVI they purified is phosphorylated? However, they have not done this. Do they know? Again, I think this is important to know and relatively simple to find out.

Response: Although we cannot definitively confirm whether our purified protein is phosphorylated, there are indications suggesting that the myosin VI sample used in our study is unlikely to be highly phosphorylated. Firstly, the majority of myosin VI molecules can form the autoinhibited conformation in the presence of ATP or its analogues (Fig. 1d). Given the reviewer’s previous claim that the phosphorylation of myosin VI may disrupt its autoinhibited state, our observation of robust autoinhibition formation of myosin VI argues against the occurring of disruptive phosphorylation. Secondly, the phosphorylation of myosin VI at specific sites, as represented in Extended Data Fig. 12, might lead to additional densities corresponding to phosphoryl groups, which, however, were not observed in our cryo-EM maps.

In extended figure 8a, an ATPase cycle is shown with arrows, that seems to suggest the cycle progresses from rigor to post rigor to prepower stroke to ADP binding. I’m afraid I don’t quite understand this, because the pre-power stroke should be ATP/ADP.Pi (so why not have ATP/ADP.Pi here) and the next step is not ‘ADP binding’ (as the ADP doesn’t bind), rather Pi is lost leaving ADP behind. Can the authors modify this diagram to make it clearer. Moreover, in the myosin field, we normally consider the cycle as starting with the pre-power stroke, not rigor. The LA wouldn’t swing back to the ADP state, as the lever position in the ADP state is after phosphate release (line 275) – so the description here seems to be a bit confusing. This logic just needs making a bit clearer, and the diagram also made clearer.

Response: Following the reviewer’s suggestion, we have modified the ATPase cycle figure. We transformed the original rectangular shape to a circled one, eliminating potential biases in understanding the sequence of different states. Furthermore, we have changed the term “ADP-binding state” to “ADP state” to avoid any misinterpretation. To further enhance clarity, we added labels for the events in the ATPase cycle, such as ATP binding, ATP hydrolysis, Pi release, and ADP release, aligning them with the labels in panel g. Meanwhile, to ensure a clear representation, we have modified the corresponding description as following: “However, as trapped in this orientation, Pi cannot be released and the LA cannot swing to adopt the ADP state (Extended Data Fig. 9g), resulting in the prolonged retention of hydrolysis products, ADP and Pi, at the active site and causing the ATPase cycle to stall”.

Line 363: The statement ‘Although the MDs of both myosin II and myosin VI are trapped in the PPS state, their LAs extends in opposite directions, due to the unique In2 sequence in

myosin VI' seems a bit odd – as the unique sequence is exactly what makes Myosin VI a 'backwards motor'. So, clearly the lever is going to be differently oriented in the PPS. Perhaps at line 40, a short sentence that explains that the insert positions the lever in such a way as to drive its movement towards the pointed end of F-actin in contrast to other myosins, would be useful background information, with a short follow up here. (Helpful for people not directly in the myosin field).

Response: As suggested, we have added the background information of “Myosin VI was characterized as the only myosin walking towards the minus ends of actin filaments, due to the unique sequence of Insert2 (In2) connecting the MD and LA in myosin VI that drives the LA movement towards the reverse direction in contrast to other myosins.” in the Introduction and revised the statement as “Although the MDs of both myosin II and myosin VI are trapped in the PPS state, their LAs extend in opposite directions, due to the unique In2 sequence in myosin VI that serves as a reverse gear for the walking direction.”.

I also don't understand the statement 'While myosin II and myosin Va employ light chain molecules to shape their hinges, and the coiled coil dimerizes hinges for further stabilization'. This seems wrong to me: the light chains are not bound to a hinge, and I'm not sure what 'dimerised' hinges means or why this might stabilise the molecule. The key difference here is that myosin V and II are dimeric, and myosin VI isn't! This statement needs removing, or changing.

Perhaps also of note here is that kinesin-1 does use a mechanism to block its active site in its shutdown state (e.g. work by Hackney and others)

Response: As suggested, we have optimized the sentence to “While myosin II and myosin Va directly employ light chain molecules for their hinge formation, and the two hinges are further stabilized through the following dimerized coiled-coil structure, myosin VI, in contrast, relies on its unique SAH and LAE for hinge stabilization in the monomeric structure.”

We acknowledge the importance of head-to-tail mediated autoinhibition in kinesin-1 (*Kaan HY, Hackney DD, Kozielski F. Science. 2011;333:883-885*). However, the paper reported that the tail in kinesin-1 dimerizes two motor domains, instead of directly blocking the active site.

Just a small comment: The use of the term 'lever arm' is incorrect, and 'lever' should be used instead: A lever is a rigid body capable of rotating on a point on itself. A lever arm is the perpendicular distance from the fulcrum of a lever to the line of action of the effort and (in physics) includes the idea of the amount of force needed to generate (or prevent) movement of an object around an axis. I realise that lever arm is generally used in the myosin field, but it's actually wrong – and should just be lever.

Response: We acknowledge the nomenclature controversy. As the reviewer's mentioned, the term “lever arm” has been generally used in the myosin field. Thus, to avoid potential confusion among readers, we have retained the well-accepted term “lever arm” in this manuscript. We hope the more discussions in the myosin field will help clarify this nomenclature issue in the future.

While I'm happy that the authors have tempered their language for the autoinhibited state to

enable diffusion, rather than conserve ATP, it would have been good if they had actually done the calculation – just how much energy does it save to have Myosin VI in an autoinhibited state?

Response: Based on the previous reports, the ATP turnover rate (V_{\max}) for wild-type myosin VI is $\sim 6 \text{ s}^{-1}\text{head}^{-1}$ (*De La Cruz EM, et al. Kinetic mechanism and regulation of myosin VI. J Biol Chem. 2001 Aug 24;276(34):32373-81.*) and the protein concentration of cellular myosin VI is $\sim 260 \text{ nM}$ (data from OpenCell database, <https://opencell.czbiohub.org>). The cellular ATP concentration in normal animal cells is several millimole (1-4 mM, *Traut TW. Physiological concentrations of purines and pyrimidines. Mol Cell Biochem. 1994 Nov 9;140(1):1-22.*) and a cell recycles all its ATP molecules about every 20-30 seconds. If we assume that all myosin VI molecules are active in a cell, they would consume $\sim 1\text{-}5\%$ of newly synthesized ATP per second, which is a proportion that cannot be simply ignored. Thus, the autoinhibition of myosin VI in cells indeed play a role in preventing unnecessary ATP consumption.